# HOTA: Hamiltonian framework for Optimal Transport Advection

## Abstract

Optimal transport (OT) has become a natural framework for guiding the probability flows. Yet, the majority of recent generative models assume trivial geometry (e.g., Euclidean) and rely on strong density-estimation assumptions, yielding trajectories that do not respect the true principles of optimality in the underlying manifold. We present Hamiltonian Optimal Transport Advection (HOTA), a Hamilton–Jacobi–Bellman based method that tackles the dual dynamical OT problem explicitly through Kantorovich potentials, enabling efficient and scalable trajectory optimization. Our approach effectively evades the need for explicit density modeling, performing even when the cost functionals are non-smooth. Empirically, HOTA outperforms all baselines in standard benchmarks, as well as in custom datasets with non-differentiable costs, both in terms of feasibility and optimality.

## 1 Introduction

*Static (Monge-Kantorovich)* optimal transport was originally considered as the main framework for comparing and finding a cost-minimizing coupling between distributions [Villani et al., 2008], while optimality was mainly measured through the boundary marginals. Development of efficient and scalable OT solvers [Cuturi, 2013, Peyré et al., 2019] popularized OT across different areas, such as generative modeling [Makkuva et al., 2020, Korotin et al., 2022, Buzun et al., 2024], computational biology [Bunne et al., 2022], graphics [Bonneel and Digne, 2023], high-energy physics [Nathan T. Suri, 2024], and reinforcement learning [Klink et al., 2022, Asadulaev et al., 2024, Bobrin et al., 2024, Rupf et al., 2025]. However, one crucial limitation of static formulation is its inability to produce non-straight paths, which completely ignores the underlying geometry of the manifold of the data. In classical OT, the underlying geometric structure is solely determined by the choice of cost function (*e.g.*, , Euclidean distance), inherently limiting the capacity for fine-grained control over the trajectories. We refer to [Montesuma et al., 2024, Pereira and Amini, 2025] for recent overview of practical applications of OT and to Villani et al. [2008], Santambrogio [2015], Peyré et al. [2019] for a formal treatment.

On the other hand, the *dynamical* optimal transport paradigm, developed by Benamou and Brenier [2000], recasts static OT as a continuous-time variational problem on the space of probability paths, effectively incorporating time variable and enabling more nuanced control over optimal trajectories (*e.g.*, , through velocity, acceleration, length, or energy over the paths). Importantly, such formulation enables one to directly operate on manifolds of non-trivial geometry, whenever the underlying space contains curvature, obstacles, or is defined through potentials. This formulation is closely connected to stochastic optimal control (SOC), where trajectories are stochastic yet must still maintain optimality, a problem class known as the generalized Schrödinger bridge (GSB) Liu et al. [2024], Bartosh et al. [2024],.

A common strategy for GSB involves solving the dual formulation via Hamilton-Jacobi-Bellman (HJB) equations, which provide a flexible and a theoretically grounded framework for deriving optimal trajectories (Liu et al. [2022], Neklyudov et al. [2024]). These methods parameterize the

cost through a Lagrangian, enforcing optimality via the preservation of kinetic energy or using other path-based penalties. While HJB-based approaches yield theoretically sound solutions, they suffer from critical drawbacks: (1) unstable optimization dynamics, leading to high-variance gradients and poor sample efficiency in high dimensions, and (2) the absence of a strict terminal distribution matching criterion, resulting in inexact couplings. Additionally, they typically require differentiable Lagrangians, restricting applicability to smooth costs only.

In the current work, we study the Generalized Schrodinger Bridge problem between two measures, where the underlying geometry is defined through potentials. We propose a new HJB-based framework that explicitly solves GSB task, resolves the learning stability problems of the previous approaches, and has theoretical guarantees. We conduct extensive empirical evaluations on existing low-dimensional physically-inspired benchmarks, as well as in the high-dimensional generative setting. In short, our contributions are as follows:

- Hamiltonian dual reformulation of dynamical OT that binds Kantorovich potentials with an HJB value function, yielding a density-free objective and providing the performance gain compared to existing works;
- Proposed approach is robust to complex geometries and works even with non-smooth cost functions as the proposed objective explicitly incorporates the potential term;
- HOTA attains state-of-the-art empirical results in a diverse set of tasks, demonstrating both better feasibility (exact marginal matching) and optimality (cost along trajectories) compared to current dynamic OT solvers.

## 2 Related work

**Diffusion Models and Matching Algorithms.** Diffusion models have emerged as powerful tools for generative modeling by prescribing the time evolution of marginal distributions. Matching algorithms, such as Action Matching (Neklyudov et al. [2024]) and Flow Matching (Lipman et al. [2023]), learn stochastic differential equations (SDEs) that align with prescribed probability paths [Blessing et al., 2025]. These methods typically assume explicit or implicit intermediate densities of the flow, whereas our approach (HOTA) optimizes a complete stochastic path from source to target distributions.

**Generalized Schrödinger Bridge.** The GSB problem extends SB by introducing state costs that penalize or reward specific trajectories (Chen et al., 2015). Prior methods for solving GSB, such as DeepGSB (Liu et al. [2022]), often relax feasibility constraints or rely on Sinkhorn-based approximations, which can lead to instability or suboptimal solutions.

A recent approach GSBM [Liu et al., 2024] follows an alternating optimization scheme: in the first stage, it learns the drift field $v_t$ while keeping the marginal distributions $\rho_t(x_t)$ fixed, using a Flow Matching-style objective. In the second stage, it updates the marginals conditioned on the boundary-coupled distribution $\rho_t(x_t \mid x_0, x_1)$, which is defined via the previously learned drift. While GSBM demonstrates strong empirical performance, it imposes two critical limitations: 1) it requires the state cost function $U(x_t)$ to be differentiable everywhere, and 2) it assumes that the conditional marginals $\rho(x_t \mid x_0, x_1)$ are Gaussian. The first constraint restricts the method's applicability to domains with smooth geometries, sometimes mitigated via interpolation [Kapusniak et al., 2024], while the second can lead to suboptimal solutions, unless $U_t$ function is not quadratic.

**Stochastic Optimal Control.** The connection between GSB and stochastic optimal control (SOC) has been explored in prior works (Theodorou et al. [2010]; Levine [2018]). SOC formulations often relax hard distributional constraints into soft terminal costs, which can introduce bias or require adversarial training (Liu et al. [2022]). Recently introduced Adjoint Matching approach [Domingo-Enrich et al., 2024a] and Stochastic Optimal Control matching (SOCM) [Domingo-Enrich et al., 2024b] address several existing limitations, but still produce highly unstable variance estimations. Our method provides a natural way to preserve the feasibility via Kantorovich potential sum.

## 3 Preliminaries

Consider stochastic process with controlled drift and diffusion:

$$\mathrm{d}x_t = v(t, x_t)\,\mathrm{d}t + \sigma(t, x_t)\,\mathrm{d}W_t \tag{1}$$

where $v : [0, 1] \times \mathbb{R}^d \to \mathbb{R}^d$ is the drift (control), $\sigma : [0, 1] \times \mathbb{R}^d \to \mathbb{R}$ is the diffusion coefficient, $W_t$ is $d$-dimensional Brownian motion. We solve the OT minimization task with marginal distributions ($\alpha$,

β) and dynamic cost functions $c(x, \mu)$ and stochastic transport mapping $\mu : \mathbb{R}^d \to \mathcal{P}(\mathbb{R}^d)$ presented in paper Korotin et al. [2022]

$$c(x, \mu) = \inf_{v(t,x):\, x_0 = x,\, x_1 \sim \mu} \int_0^1 \mathbb{E}\mathcal{L}(t, x_t, v_t) dt, \quad \mathcal{L}(t, x_t, v_t) = \frac{\|v_t\|^2}{2} + U(x_t). \quad (2)$$

This problem is also known as generalized Schrödinger bridge (GSB). It is an extension of the classical Schrödinger Bridge (SB) problem, which is a distribution-matching task seeking a diffusion model that transports an initial distribution $\alpha$ to a target distribution $\beta$. While the standard SB minimizes the kinetic energy ($L^2$ cost in OT), the GSB introduces additional flexibility by incorporating a state cost $U(x_t)$, allowing for more general optimality conditions beyond just kinetic energy minimization. The standard SB's reliance on kinetic energy (Euclidean cost) may not be ideal for all applications (e.g., image spaces, where distance may not be meaningful). Many scientific domains (population modeling, robotics, molecular dynamics) require richer optimality conditions, which GSB accommodates via $U(x_t)$. The potential term usually characterizes the geometry of the space. But in addition, we can also include some physical properties of the flow, *e.g.*, entropic penalty or "mean-field" interaction [Liu et al., 2022]. Thus, the optimal trajectories are curved to avoid regions with high values of $U(x_t)$.

Neural networks can effectively solve high-dimensional Optimal Transport (OT) problems by learning the Kantorovich potentials, which maximizes the dual objective (Korotin et al. [2022], Buzun et al. [2024]). It is shown in Villani et al. [2008] (Theorem 5.10) that OT task is equivalent to the maximization of the Kantorovich potentials sum:

$$\sup_{g \in L_1(\beta)} \Big[ \mathbb{E}_\alpha[g^c(x)] + \mathbb{E}_\beta[g(y)] \Big], \quad (3)$$

where $g^c$ denotes $c$-conjugate transform of the potential $g$:

$$g^c(x) = \inf_{\mu(x):\, \mathbb{R}^d \to \mathcal{P}(\mathbb{R}^d)} \mathbb{E}_{y \sim \mu(x)} \Big[ c(x, \mu) - g(y) \Big]. \quad (4)$$

Here $\mu(x)$ is the stochastic transport mapping, and in our notation it is the final distribution of the stochastic process $x_1$ under condition that $x_0 = x$. The marginality requirement of the final distribution of $x_1$ (which must correspond to $\beta$) is ensured by the potential difference $\mathbb{E}_\beta g(y)$ and $\mathbb{E}_\alpha \mathbb{E}_{y \sim \mu(x)} g(y)$, which tends to infinity otherwise.

But unlike classical OT, we need to minimize the cost throughout the trajectory $x_t$, $t \in [0, 1]$ with the following objective

$$g^c(x) = \inf_{v(x,t)} \mathbb{E} \left[ \int_0^1 \left( \frac{\|v(t, x_t)\|^2}{2} + U(x_t) \right) \mathrm{d}t - g(x_1) \,\Big|\, x_0 = x \right]. \quad (5)$$

In the last expression, we have united infimums by $\mu(x)$ and control $v(t, x)$ and as a sequence have removed the right side condition $x_1 \sim \mu(x)$. Based on dynamic programming approach, define the value function. For any $0 \le t \le 1$, the value function satisfies:

$$s(t, x) = \inf_{x_t} \mathbb{E} \left[ \int_t^1 \left( \frac{\|v(t, x_t)\|^2}{2} + U(x_t) \right) \mathrm{d}t - g(x_1) \,\Big|\, x_t = x \right], \quad (6)$$

such that our objective equals $s(0, x)$ and the boundary condition at time point $t = 1$ is

$$\forall x \in \mathbb{R}^d : s(1, x) = -g(x).$$

Function $s(t, x)$ solves the Hamilton-Jacobi-Bellman (HJB) differential equation and it in turn allows us to find the conjugate potential $g^c$ (4).

$$-\partial_t s(t, x) = \inf_v \{ v^T \nabla_x s(t, x) + \mathcal{L}(t, x, v) \} + \frac{\sigma^2}{2} \mathrm{tr} \{ \nabla^2 s(t, x) \}. \quad (7)$$

Representation of the Lagrange function as a sum of kinetic and potential energy allows us to find the minimum in velocity ($v$) in explicit form, such that $v_t = -\nabla_x s(t, x_t)$. Together with potential optimization (3), we obtain the final GSB objective in dual Kantorovich form. We provide a detailed proof in Section 6.

**Theorem 1** (Dual GSB problem). *Given distributions $\alpha, \beta \in \mathcal{P}(\mathbb{R}^d)$ and stochastic dynamics (1) with cost functional (2), the dynamic optimal transport problem admits the following formulation:*

$$\max_{s(1,\cdot)\in L_1(\beta)} \{\mathbb{E}_\alpha \, s(1, x_1) - \mathbb{E}_\beta \, s(1, y)\} \tag{8}$$

*where $s(t, x) \in C^{1,2}([0, 1] \times \mathbb{R}^d)$ and satisfies HJB PDE $\forall t \in [0, 1]$ and $\forall x \in \mathbb{R}^d$*

$$-\partial_t s(t, x) = -\frac{1}{2}\|\nabla_x s(t, x)\|^2 + U(x) + \frac{\sigma^2}{2}tr\{\nabla^2 s(t, x)\}. \tag{9}$$

The first expression in Theorem 1 plays the role of a discriminator and guarantees matching the target distribution $\beta$, and the second one is responsible for the optimality of trajectories. For the HJB equation to have a unique solution (in the viscosity sense), we require *coercivity* (Theorem 4.1 [Fleeting and Soner, 2006]) of the Hamiltonian for some constants $C_1 > 0$ and $C_2 \geq 0$

$$H(x, \nabla s, \nabla^2 s) = \frac{1}{2}\|\nabla_x s\|^2 - U(x) - \frac{\sigma^2}{2}tr\{\nabla^2 s\} \tag{10}$$

$$\geq C_1(\|\nabla s\|) - C_2(1 + \|x\| + \|\nabla^2 s\|) \tag{11}$$

The term $\|\nabla_x s\|^2$ dominates for large values, so in case $U(x)$ is bounded and $\sigma > 0$ the solution is unique. By means of the optimized function $s(t, x)$ we can generate the OT trajectories using Euler-Maruyama algorithm:

$$x_{t+\Delta t} = -\nabla_x s(t, x_t)\Delta t + \sigma \Delta W, \quad x_0 \sim \alpha. \tag{12}$$

Unlike most other methods, here we do not need to model the intermediate density of the $x_t$ ($t \in (0, 1)$) distribution, which greatly simplifies the learning process, but we need to store the generation history in a replay buffer for more stable HJB optimization in high-dimensional spaces.

## 4 Method

To find a stable and balanced solution $s(t, x)$ for the given dynamic OT problem (1), we can follow a composite approach that combines optimal control (via HJB PDE constraints) and RL techniques (policy-based trajectory optimization). We approximate the value function using a parametric model $s_\theta(t, x)$. We have to maximize the potential matching functional (8) subject to the constraint that $s_\theta(t, x)$ satisfies the HJB PDE. For that divide the time interval $[0, 1]$ into $T$ time steps and simulate $n$ trajectories $\{t_0^k, x_0^k, \ldots, t_T^k, x_T^k\}_{k=1}^n$ using initial $\alpha$ distribution and Euler-Maruyama method (12). Sample also $n$ points $y_k$ from the target distribution $\beta$ and compute the potential matching loss as

$$L_{\text{pot}}(s_\theta) = \frac{1}{n}\sum_{k=1}^n s_\theta(1, x_T^k) - \frac{1}{n}\sum_{k=1}^n s_\theta(1, y^k). \tag{13}$$

The HJB PDE must hold for all $t \in [0, 1]$ and $x \in \mathbb{R}^d$, but in practice, for more effective training, the training data should be sampled in the region of the flow (trajectories) concentration (according to Liu et al. [2022]). We enforce this by linear interpolation between datasets from $\alpha$ and $\beta$ as a rough estimation of the flow region and subsequently use the replay buffer $\mathcal{B}$ to collect points from the previously obtained trajectories. Using data samples $\{t^k, x^k\}_{k=1}^n$ from $\mathcal{B}$ or the linear interpolation we compute HJB residual loss as

$$L_{\text{hjb}}(s_\theta, \overline{s}) = \frac{1}{n}\sum_{k=1}^n \left(\frac{\partial s_\theta^k}{\partial t} - \frac{1}{2}\|\nabla_x \overline{s}^k\|^2 + U(x^k) + \frac{\sigma^2}{2}tr\{\nabla^2 \overline{s}^k\} + \lambda_a\|a^k\|\right)^2 \tag{14}$$

$$+ \frac{1}{n}\sum_{k=1}^n \left(\frac{\partial \overline{s}^k}{\partial t} - \frac{1}{2}\|\nabla_x s_\theta^k\|^2 + U(x^k) + \frac{\sigma^2}{2}tr\{\nabla^2 s_\theta^k\} + \lambda_a\|a^k\|\right)^2, \tag{15}$$

where $s_\theta^k = s_\theta(t^k, x^k)$, $\overline{s}^k = \overline{s}(t^k, x^k)$ denotes the target model with EMA parameters, $a^k$ is angular acceleration defined as

$$a^k = \frac{d}{dt}\frac{\nabla s_\theta(t^k, x^k)}{\|\nabla s_\theta(t^k, x^k)\|}. \tag{16}$$

The angular acceleration with coefficient $\lambda_a$ forces the straightening of the trajectories (optionally). We divide the model into $s_\theta$ and $\overline{s}$ as it usually done in RL methods to make the optimization problem more similar to regression.

In the result, our model is trained on two criteria ($L_{\text{pot}}$ and $L_{\text{hjb}}$) simultaneously and to balance both impacts we scale the gradients of the hjb-loss and sum it with the pot-loss:

$$\nabla_\theta L_{\text{pot}}(s_\theta) + \lambda_{\text{hjb}}\text{EMA}\left(\frac{\|\nabla_\theta L_{\text{pot}}(s_\theta)\|}{\|\nabla_\theta L_{\text{hjb}}(s_\theta, \overline{s})\|}\right)\nabla_\theta L_{\text{hjb}}(s_\theta, \overline{s}). \tag{17}$$

The complete method is implemented as shown in Algorithm 1. It effectively combines the theoretical guarantees of optimal transport with the flexibility of neural network approximations, while maintaining numerical stability through careful gradient management. The adaptive balancing of the potential matching and HJB residual losses ensures stable convergence to a solution that satisfies both the optimality conditions and the boundary constraints.

---

**Algorithm 1** HOTA: Hamiltonian framework for Optimal Transport Advection

1: **Input**: value model $s_\theta$, model optimizer s_opt, distributions $\alpha$ and $\beta$, potential function $U(x)$, diffusion coefficient $\sigma$.
2: **Hyperparameters**: train steps $N$, iterpolation sample steps $N_0$, temporal discretization $T$, batch size $n$, hjb-loss weight $\lambda_{\text{hjb}}$, acceleration coefficient $\lambda_a$, learning rate lr, gradients scale EMA coefficient $\tau$.
3: **Initialize** target model $\overline{s}$; replay buffer $\mathcal{B} \leftarrow \emptyset$; gradients scale $\alpha \leftarrow 1.0$
4: **for** iteration $i = 1$ to $N$ **do**
5:     Sample train data $\{x_0^k\}_{k=1}^n \sim \alpha$; $\{y^k\}_{k=1}^n \sim \beta$
6:     **if** $i < N_0$ **then**
7:         Sample times $\{t^k\}_{k=1}^n \sim U(0,1)$
8:         For $1 \leq k \leq n$ set $x^k = x_0^k \cdot (1 - t^k) + y^k \cdot t^k$
9:     **else**
10:         Sample $\{t^k, x^k\}_{k=1}^n \sim \mathcal{B}$
11:     **end if**
12:     Generate $n$ trajectories $\{t_0^k, x_0^k, \ldots, t_T^k, x_T^k\}_{k=1}^n$ using current policy $v_t = -\nabla s(t, x)$
13:     Add the 1-st trajectory $\{t_0^0, x_0^0, \ldots, t_T^0, x_T^0\}$ to $\mathcal{B}$
14:     **Compute gradients:**
15:     $g_{\text{hjb}} = \nabla_\theta L_{\text{hjb}}(s_\theta, \overline{s}, \{t^k, x^k\}_{k=1}^n)$
16:     $g_{\text{pot}} = \nabla_\theta L_{\text{pot}}(s_\theta, \{x_T^k\}_{k=1}^n, \{y^k\}_{k=1}^n)$
17:     **Update Parameters:**
18:     Compute norms $G_{\text{hjb}} = \|g_{\text{hjb}}\|_2$ and $G_{\text{pot}} = \|g_{\text{pot}}\|_2$
19:     EMA update of gradients scale $\alpha = \tau G_{\text{pot}}/G_{\text{hjb}} + (1 - \tau)\alpha$
20:     Sum the gradients $g = g_{\text{pot}} + \lambda_{\text{hjb}}\, \alpha\, g_{\text{hjb}}$
21:     Update model parameters $\theta$ with s_opt($g$)
22:     EMA update of target model $\overline{s}$
23: **end for**

---

## 5 Experiments

In this section, we evaluate our method on a series of distribution matching tasks with non-trivial geometries. In Section 5.2, we compare HOTA with state-of-the-art baselines, demonstrating its superior performance on both standard benchmarks including datasets with almost non-differentiable potentials. In Section 5.3, we demonstrate the scalability of our approach by showcasing its effectiveness in high-dimensional settings. Finally, in Section 5.4, we ablate key components of our method.

### 5.1 Experimental Setup

**Evaluation** We assess performance using two metrics: *feasibility* and *optimality*. Feasibility reflects how well the method matches the target distribution, evaluated via Wasserstein distance with squared Euclidean cost ($W_2(T_\#\alpha, \beta)$), where the transport mapping $T$ uses optimized value function $s_\theta$ and samples $x_1$ by procedure (12). Optimality measures the quality of the resulting mapping, estimated through the integral trajectory cost: $\int_0^1 \left[\mathbb{E}_{\rho_t} \frac{\|v_t(x_t)\|^2}{2} + U(x_t)\right] dt$, where $x_t$ follows (1).

**Network** In all our experiments, we employ a simple MLP augmented with Fourier feature encoding of the time component. For general time embeddings of the form $\text{emb}(t) = \sin(f \cdot t + \varphi)$, the time

derivative is given by $\partial_t \mathrm{emb}(t) = f \cdot \cos(f \cdot t + \varphi)$. As the frequency $f$ increases, the magnitude of this derivative also grows, potentially leading to numerical instability—especially when the time derivative of the network is explicitly involved in the objective. This issue has been previously discussed in Lu and Song [2024]. To address this, we restrict the frequency range to $[1, 20]$ and normalize the resulting Fourier features by dividing by the corresponding frequencies.

**Baselines** We use source code from GSBM repository for running it in our experiments on BabyMaze, Slit and Box datasets. Other results were taken from the original papers Liu et al. [2024], Pooladian et al. [2024] where dataset were previously introduced.

All experiments are conducted on a GeForce RTX 3090 GPU and take less than ten minutes for training. Additional experimental details are provided in Appendix A.

## 5.2 Comparative Evaluation on Two-Dimensional Data

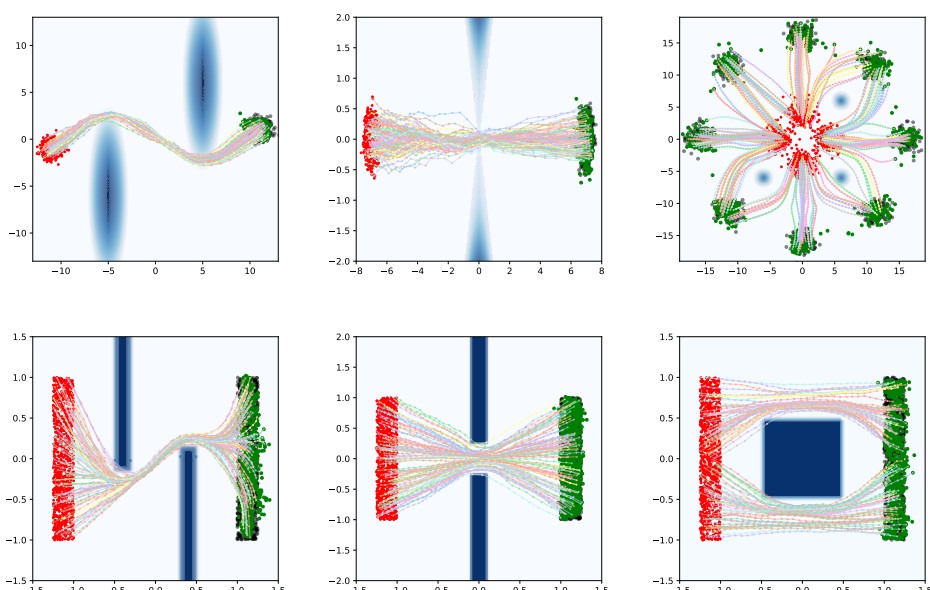

Figure 1: Evaluation of HOTA method on smooth (top) and non-smooth datasets (bottom): Stunnel, Vneck, GMM, BabyMaze, Slit, Box. Blue regions indicate high values of potential $U(x)$. Distributions $\alpha$ (red), $\beta$ (black) and the mapped $T_{\#}\alpha$ (green).

In this section, we compare our method to previous state-of-the-art approaches on the standard benchmarks including datasets that feature almost non-differentiable potential functions. Visualizations of the datasets are provided in Figure 1. The first three datasets—Stunnel, Vneck, and GMM—are adopted from Liu et al. [2024]. These benchmarks incorporate state cost functions $U(x_t)$ that encourage the optimal solution to respect complex geometric constraints. Each dataset is designed to highlight specific capabilities of the evaluated algorithms. *Stunnel* assesses whether a method can capture drift fields that undergo rapid and localized changes. *Vneck* evaluates the ability to model drift that compresses and expands the support of marginal distributions. *GMM* tests whether the method can disambiguate closely situated points and assign them to distinct trajectories. The remaining datasets—BabyMaze, Slit, and Box (Pooladian et al. [2024])—are constructed using similar underlying principles but pose additional difficulties due to the presence of almost non-differentiable state cost functions. A summary of the quantitative results across all datasets is provided in Table 1. Our method, HOTA, consistently outperforms existing approaches in terms of both feasibility and optimality. In particular, HOTA achieves a substantial performance gain on the GMM dataset, which may refer to its superior capability in trajectory separation for closely situated points.

## 5.3 Scalability to High-Dimensional Spaces

In this section, we test the scalability of our method, demonstrating its stable performance in higher-dimensional settings. For this purpose, we use *Sphere* datasets parameterized by data dimensionality

Table 1: Quantitative comparison between recent state-of-the-art methods and our approach, HOTA. Performance is evaluated using two criteria: *Feasibility* (how well the target distribution is covered) and *Optimality* (efficiency of the learned mapping). Our method consistently outperforms existing approaches, with significantly better results in certain tasks, such as GMM. N/A cells indicate that original authors of particular method did not include results for those tasks. The mean and the standard deviations of our method are computed across 5 different seeds. Best values are highlighted by bold font (lower is better). Gray values correspond to the method's divergence.

| | Feasibility $W_2(T_\#(\alpha), \beta)$ | | | Optimality (integral cost) | | |
| --- | --- | --- | --- | --- | --- | --- |
| | Stunnel | Vneck | GMM | Stunnel | Vneck | GMM |
| NLSB | 30.54 | 0.02 | 67.76 | 207.06 | 147.85 | 4202.71 |
| GSBM | 0.03 | 0.01 | 4.13 | 460.88 | 155.53 | 229.12 |
| **HOTA** | **0.006**$_{\pm 0.003}$ | **0.002**$_{\pm 0.001}$ | **0.19**$_{\pm 0.05}$ | **320.90**$_{\pm 12.5}$ | **115.09**$_{\pm 8.9}$ | **80.44**$_{\pm 2.6}$ |
| | BabyMaze | Slit | Box | BabyMaze | Slit | Box |
| NLSB | $> 1$ | 0.013 | 0.024 | N/A | N/A | N/A |
| NLOT | $> 1$ | 0.013 | 0.016 | N/A | N/A | N/A |
| GSBM | 0.01 | 0.01 | 0.02 | 6.5 | 4.9 | 3.8 |
| **HOTA** | **0.004**$_{\pm 0.003}$ | **0.0004**$_{\pm 0.0001}$ | **0.002**$_{\pm 0.001}$ | **4.87**$_{\pm 0.14}$ | **3.06**$_{\pm 0.09}$ | **2.84**$_{\pm 0.11}$ |

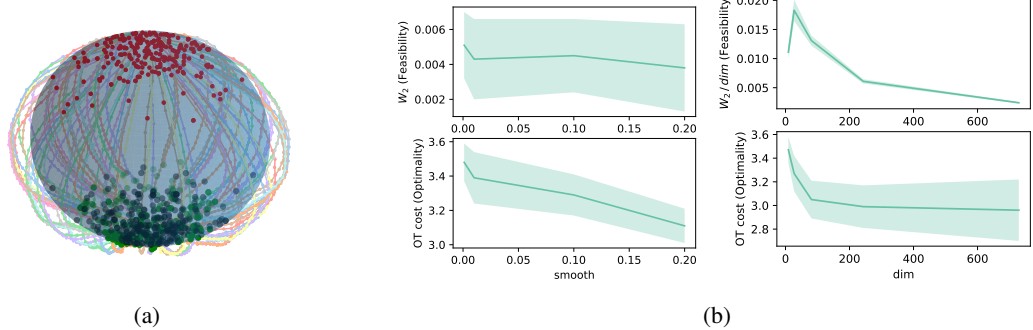

(a)             (b)

Figure 2: (a) Visualization of *Sphere* dataset for $N = 3$. (b) *Feasibility* and *Optimality* trends with respect to $3D$ unit sphere smoothness (left) and unit sphere dimensionality (right). Our method maintains robust performance across both non-differentiable potentials and high-dimensional settings.

$N$. Specifically, we define an $N$-dimensional unit sphere as a potential barrier inducing corresponding state cost function $U(x_t)$. The source and target distributions are samples from a standard distribution located at the poles, projected onto the unit sphere. The three-dimensional case is visualized in Figure 2a. The performance of our method across varying data dimensions is shown in Figure 2b (right). Notably, HOTA demonstrates robust and stable performance as the dimensionality $N$ increases.

### 5.4 Ablation study

Table 2 presents comparison of the full HOTA model against variants without the replay buffer $\mathcal{B}$ that stores simulation history or the adaptive gradient balancing by means of $\alpha$ (17), evaluating as previously feasibility and optimality metrics across Stunnel, Vneck, and GMM datasets. The full HOTA achieves strong metric scores, while removing the buffer severely degrades feasibility in Vneck and GMM and increases costs in Stunnel. Disabling gradient balancing harms feasibility in Stunnel and GMM. The results highlight the buffer's critical role in maintaining feasibility and the nuanced trade-offs between gradient balancing and transport efficiency across different scenarios.

Additionally we have evaluated the influence of acceleration term $\lambda_a \|a\|$ used in loss $L_{\text{hjb}}$ depending on $\lambda_a$ (Figure 3). It performs the function of straightening trajectories by penalizing the change in angular velocity. It follows from the results that increasing $\lambda_a$ improves the optimality of the transportation trajectories while introducing a small bias in the matching of the target distribution $\beta$, which is reflected in the feasibility metric. In the GMM task, due to the specificity of the dataset

and the divergence of trajectories in different directions, a small penalization of acceleration also improves feasibility.

Table 2: Comparison of HOTA method against variants without the replay buffer $\mathcal{B}$ and the adaptive gradient balancing. Best values are highlighted by bold font (lower is better). Gray values correspond to the method's divergence.

| | Feasibility $W_2(T_\#(\alpha), \beta)$ | | | Optimality (integral cost) | | |
|---|---|---|---|---|---|---|
| | Stunnel | Vneck | GMM | Stunnel | Vneck | GMM |
| HOTA | **0.006** | **0.002** | **0.19** | **320.90** | 115.09 | **80.44** |
| HOTA w/o buffer | 0.076 | 16.47 | 1.248 | 706.89 | 82.49 | 121.6 |
| HOTA w/o grad. balancing | 3.60 | 0.026 | 2.64 | 325.22 | **109.25** | 72.77 |

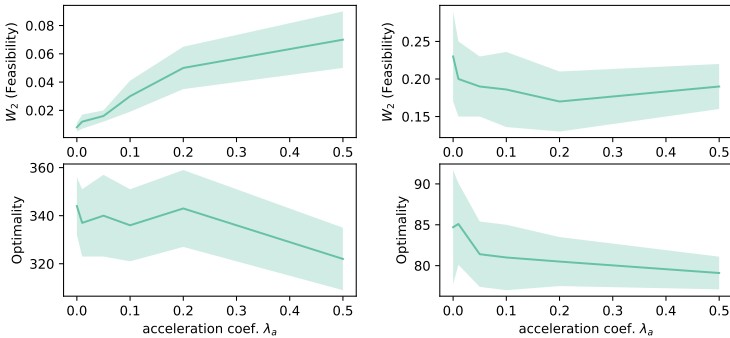

Figure 3: Impact of acceleration coefficient $\lambda_a$. Left: Stunnel, right: GMM datasets.

## 6  Proof of Theorem 1 (Dual Formulation of GSB)

We prove in the **first step** the equivalence between the GSB (stochastic control formulation) and its dual formulation using Kantorovich-style duality. Remind that we consider the stochastic process $x_t$ (1) with conditions $x_0 \sim \alpha$, $x_1 \sim \beta$, control function $v(t, x_t)$, and Brownian motion $\sigma(t, x_t)dW_t$. The **primary problem** of GSB optimization is:

$$\inf_{v(x,t)} \mathbb{E}\left[\int_0^1 \mathcal{L}(t, x_t, v_t)dt\right] \quad \text{s.t.} \quad x_0 \sim \alpha, \; x_1 \sim \beta, \tag{18}$$

where in the particular case $\mathcal{L}(t, x, v) = v^2/2 + U(x)$. Since the stochastic process $x_t$ starts from $x_0 \sim \alpha$ the primal problem is equivalent to:

$$\inf_{v(t,x)} \left( \mathbb{E}\left[\int_0^1 \mathcal{L}(t, x_t, v_t)dt\right] + \sup_{g \in L_1(\beta)} \left(-\mathbb{E}[g(x_1)] + \mathbb{E}_\beta[g(y)]\right) \right), \tag{19}$$

where the supremum over $g$ enforces the constraint $x_1 \sim \beta$ (via Lagrange duality). Rewrite the Lagrangian problem as

$$\inf_{v(t,x)} \sup_{g \in L_1(\beta)} \left( \mathbb{E}\left[\int_0^1 \mathcal{L}(t, x_t, v_t)dt - g(x_1)\right] + \mathbb{E}_\beta[g(y)] \right). \tag{20}$$

Assuming strong duality holds under mild regularity conditions (e.g., $\mathcal{L}$ convex in $v$, $\alpha, \beta$ absolutely continuous), we swap $\inf$ and $\sup$:

$$\sup_{g \in L_1(\beta)} \left( \inf_{v(t,x)} \mathbb{E}\left[\int_0^1 \mathcal{L}(t, x_t, v_t)dt - g(x_1)\right] + \mathbb{E}_\beta[g(y)] \right). \tag{21}$$

Note that since the optimal $v^*(t, x)$ is Markovian (depends only on current time $t$ and state $x$) and does not depend on the initial distribution $\alpha$ it holds that

$$\mathbb{E}\left[\int_0^1 \mathcal{L}(t, x_t, v_t^*)dt - g(x_1)\right] = \mathbb{E}_{x \sim \alpha} \mathbb{E}\left[\int_0^1 \mathcal{L}(t, x_t, v_t^*)dt - g(x_1) \,\Big|\, x_0 = x\right]. \tag{22}$$

Buy the definition of $c$-conjugate transform (5):

$$\mathbb{E}_{x\sim\alpha}\mathbb{E}\left[\int_0^1 \mathcal{L}(t,x_t,v_t^*)dt - g(x_1)\,\bigg|\, x_0 = x\right] = \mathbb{E}_{x\sim\alpha}g^c(x). \tag{23}$$

Thus, the **dual problem** becomes: $\sup_{g\in L_1(\beta)}\left(\mathbb{E}_\alpha[g^c(x)] + \mathbb{E}_\beta[g(y)]\right)$. In the **second step** find the optimal control solution $v^*(t,x)$ by means of dynamic programming principle. Define the value function $s(t,x)$ that for any $0 \le t \le \tau \le 1$ satisfies:

$$s(t,x) = \inf_{v(t,x)}\mathbb{E}\left[\int_t^\tau \mathcal{L}(z,x_z,v_z)dz + s(\tau,x_\tau)\,\bigg|\, x_t = x\right]. \tag{24}$$

Applying Ito's formula to $s(\tau,x_\tau)$ we obtain that

$$ds(\tau,x_\tau) = \partial_\tau s\,\mathrm{d}\tau + \nabla s \cdot \mathrm{d}x_\tau + \frac{1}{2}\mathrm{tr}(\sigma^2\nabla^2 s)\,\mathrm{d}\tau \tag{25}$$

$$= \left(\partial_\tau s + \nabla s^T v_\tau + \frac{1}{2}\mathrm{tr}(\sigma^2\nabla^2 s)\right)\mathrm{d}\tau + \nabla s^T\sigma\,\mathrm{d}W_s. \tag{26}$$

Consider the evolution of the value between times $t$ and $\tau$:

$$s(\tau,x_\tau) - s(t,x_t) = \int_t^\tau\left(\partial_z s + \nabla s \cdot v_z + \frac{1}{2}\mathrm{tr}(\sigma^2\nabla^2 s)\right)\mathrm{d}z + \int_t^\tau \nabla s^T\sigma\,\mathrm{d}W. \tag{27}$$

Basing on the martingale property of Ito integrals ($\mathbb{E}[\int \nabla s \cdot \sigma\,\mathrm{d}W | x_t = x] = 0$) it holds that

$$\mathbb{E}[s(\tau,x_\tau)|x_t = x] = s(t,x) + \mathbb{E}\left[\int_t^\tau\left(\partial_z s + \nabla s \cdot v_z + \frac{1}{2}\mathrm{tr}(\sigma^2\nabla^2 s)\right)\mathrm{d}z\right]. \tag{28}$$

Substitute back into dynamic programming and plug the last expression into the equation (24):

$$s(t,x) = \inf_{v(t,x)}\mathbb{E}\left[\int_t^\tau \mathcal{L}(z,x_z,v_z)\,\mathrm{d}z + s(t,x) + \int_t^\tau\left(\partial_z s + \nabla s^T v_\tau + \frac{1}{2}\mathrm{tr}(\sigma^2\nabla^2 s)\right)\mathrm{d}z\right]. \tag{29}$$

Cancel $s(t,x)$ from both sides and divide by $(\tau - t)$:

$$0 = \inf_{v(s,t)}\frac{1}{\tau - t}\mathbb{E}\left[\int_t^\tau\left(\mathcal{L}(z,x_z,v_z) + \partial_z s + \nabla s^T v_z + \frac{1}{2}\mathrm{tr}(\sigma^2\nabla^2 s)\right)\mathrm{d}z\right]. \tag{30}$$

Take limit $\tau \downarrow t$ to derive the HJB equation for a general Lagrangian $\mathcal{L}$

$$0 = \inf_v\left\{\mathcal{L}(t,x,v) + \partial_t s + \nabla s^T v + \frac{1}{2}\mathrm{tr}(\sigma^2\nabla^2 s)\right\}. \tag{31}$$

Identify optimal control for the particular $\mathcal{L}(t,x,v) = v^2/2 + U(x)$. The infimum is attained when $v^* = -\nabla s$, yielding the final result of Theorem 1.

## 7 Limitations and Future Work

While HOTA exhibits strong and robust performance, we observed sensitivity to certain network design choices—particularly the Fourier feature encoding of time, a commonly used technique in models that estimate ODE drifts. Additionally, because the value function in our framework must simultaneously support optimal control estimation and serve as a Kantorovich potential, it requires a network architecture capable of aggregating rich temporal and spatial information. The use of a simple MLP, while effective, may not be optimal from an optimization standpoint. Incorporating architectures with stronger inductive biases could further enhance performance. These considerations lie beyond the scope of this work, but we believe they offer promising directions for future research.

## 8 Conclusion

In this work, we introduced HOTA, a new OT method based on the Hamilton–Jacobi–Bellman (HJB) framework for solving the Generalized Schrödinger Bridge problem. We demonstrated that HOTA consistently outperforms recent state-of-the-art methods on standard benchmarks and scales effectively to high-dimensional settings. Remarkably, it works for non-smooth potentials and with non-differentiable cost functions, yielding robust performance gain in terms of strictly defined concepts of feasibility and optimality.

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

## A   Additional Experimental Details

**Hyperparameters** Table 3 summarizes the hyperparameters used for each dataset presented in the paper. Note that the Sphere datasets, which are parameterized by data dimensionality, share all hyperparameters except for the potential weight, which may take value 10 for the low dimensions are 30 for high ones.

Table 3: Hyperparameters used for each dataset presented in the paper.

| Hyperparameter | Stunnel | Vneck | GMM | BabyMaze | Slit | Box | Sphere |
|---|---|---|---|---|---|---|---|
| MLP hidden layers | | | | $[512, 512, 512, 1]$ | | | |
| Fourier frequencies | | | | $\{1,\ldots,20\}$ | | | |
| optimizer | | | Adam with cosine annealing ($\alpha = 1\text{e-}2$) | | | | |
| initial learning rate | | | | $5 \times 10^{-4}$ | | | |
| Adam $[\beta_1, \beta_2]$ | | | | $[0.9, 0.99]$ | | | |
| # training iterations | | | | 70000 | | | |
| batch size | | | | 1024 | | | |
| EMA decay, $\tau$ | | | | 0.9 | | | |
| # control steps | | | | 30 | | | |
| diffusion coef., $\sigma$ | 0.3 | 0.2 | 0.1 | 0.03 | 0.05 | 0.03 | 0.01 |
| control weight, $\lambda_a$ | 1.0 | 2.0 | 0.7 | 0.5 | 2.0 | 0.3 | 0.4 |
| acc. weight, $\lambda_a$ | 0.0001 | 0.001 | 0.2 | 0.05 | 0.001 | 0.01 | 0 |
| potential weight | 25 | 1000 | 25 | 10 | 30 | 700 | $\{10, 30\}$ |

