# OpenReview forum: "HOTA: Hamiltonian framework for Optimal Transport Advection"
_NeurIPS.cc/2025/Conference — Submitted to NeurIPS 2025_

### Official Review · Reviewer_N62o · 2025-06-21

**Clarity:** 2
**Significance:** 2
**Originality:** 3
**Rating:** 3
**Confidence:** 3

**Summary:**

This work proposes a method to learn trajectories between a source and a target distribution under a Lagrangian cost which might not be differentiable. The authors propose to rewrite the dynamic optimal transport problem with such cost using a formulation based on the maximization of a potential function with respect to a value function, which is approximated using a neural network, and under the constraing that it must satisfies the HJB equation. The method is shown to work better than several baselines on different benchmarks, and is shown to be scalable to high-dimensional data, and ablation studies are performed for the different choices of the algorithm.

**Questions:**

In Section 3, equation (2) is referred to as the dynamic cost of an OT problem. But this OT problem is not introduced before, and the cost is not in a standard form. This is a bit confusing. Is it the cost of the weak OT problem? Same question for equations (3) and (4). It seems to correspond to the weak OT problem. If it is the case, I do not think Theorem 5.10 of (Villani, 2008) is the right reference for the dual given in equation (3).

The Hamilton-Jacobi-Bellman equation is not introduced before equation (7), and I do not find it very clear why the function $s(t,x)$ solves it?

Why OT trajectories can be generated by equation (12)?

What are EMA parameters, and target models in Section 4?

The experiments focus mainly on toy datasets. Even the experiment in higher dimension go only to dimension 700 on data simulated on a sphere. Would it be possible to also do experiments on real data such as images? Are there real case scenario where there would be an interest to use non differentiable potential?

Since the method can be used with general costs, it could also be a nice addition to perform metric learning as in [1,2,3].


Typos:
- In equation (6), therer are 2 parameters call $t$.
- Line 115: "and as a sequence": consequence?
- Line 116-117: "Based on dynamic programming approach, define the value function. For"
- Equation (11): miss a point


[1] Scarvelis, C., & Solomon, J. (2023, May). Riemannian metric learning via optimal transport. In International Conference on Learning Representations.

[2] Pooladian, A. A., Domingo-Enrich, C., Chen, R. T., & Amos, B. (2024). Neural optimal transport with lagrangian costs. arXiv preprint arXiv:2406.00288.

[3] Howard, S., Deligiannidis, G., Rebeschini, P., & Thornton, J. (2024). Differentiable cost-parameterized monge map estimators. arXiv preprint arXiv:2406.08399.

**Ethical Concerns:**

["NO or VERY MINOR ethics concerns only"]

**Final Justification:**

While this work is interesting, I believe that it is not very clear and that the experiments could be improved. The rebuttal was pretty good at clarifying the unclear points, but I think there is a lot of things to be changed, which would require another round of review. Thus, I am leaning towards rejection.

**Limitations:**

yes

**Quality:**

3

**Strengths And Weaknesses:**

Learning trajectories between two distribution with respect to a general cost is an important problem which received lots of attention recently. This work tackles this task successfully using a composite approach combining optimal control and RL. Moreover, the method is shown to work on several datasets and compared to different baselines. Nonetheless, I think some elements of the framework could be clarified, and experiments could be improved to real problems

**Strengths**:
- A method allowing to compute trajectories between two distributions while being optimal relative to pretty general costs.
- The method is based on an original formulation of the problem, and combines optimal control and RL elements.
- The method is compared to baselines on standard datasets.
- Ablations on some choices of the algorithms are provided.

**Weaknesses**:
- I found that Section 3, where the main theoretical results to understand the method are described, was not very clear.
- In the description of the method, there are some elements that are not clear. In particular, a target model $\bar{s}^k$ appears in Section 4 as well as "EMA" parameters. But they are not introduced before.
- The experiments are on toy problems.

---

> ### Author Rebuttal · Authors · 2025-07-31
>
> **Questions**
>
> **Q1**: In Section 3, equation (2) is referred to as the dynamic cost of an OT problem.  Is it the cost of the weak OT problem?
>
> You are absolutely right. Thank you for your question. The cost function presented in equations (2)--(4) corresponds to a weak
> optimal transport (WOT) formulation, where the cost depends on
> probability measures conditioned on source points, rather than the
> classical OT cost that depends only on pairs of points.
>
> Therefore, classical duality results such as Theorem 5.10 in Villani
> (2008), which address standard OT costs, are not directly applicable
> here. Instead, the appropriate theoretical foundation comes from the
> recent widely used work specialized on weak OT (e.g., Gozlan et al.,
> 2017; Kantorovich Duality for General Transport Costs and Applications),
> which establishes duality and existence results for these generalized
> dynamic cost settings. Namely, it shows, that given a measurable space
> $X$, family of distributions $\mathcal{P}(X)$ on it, weak cost
> $c: X \times \mathcal{P}(X) \rightarrow [0, \infty]$, source
> distribution $\alpha \in \mathcal{P}(X)$, target distribution
> $\beta \in \mathcal{P}(X)$, under mild conditions, the following primal
> problem
> $$\inf_{\pi \in \Pi(\alpha, \beta)} \int c(x, \mu_x) \alpha(dx),$$ is
> equivalent to the the dual formulation:
> $$
> \sup_{g}
>     \{
>         \int g^c(x) \alpha(dx) + \int g(y) \beta(dy)
>     \},
> $$
> where
> $$g^c(x) = \inf_{\mu \in \mathcal{P}(X)}
>     \{
>         \int - g(y) \mu(dy) + c(x, \mu)
>     \}.
> $$
> We acknowledge that citing Villani (2008) in this context may cause
> confusion, and we intend to clarify this by referencing the relevant
> weak OT literature in our revision.
>
> **Q2** : *The Hamilton-Jacobi-Bellman equation is not introduced before
> equation (7), and I do not find it very clear why the function $s(t, x)$
> solves it?*
>
> **Answer**
>
> The fact that $s(t, x)$ solves the Hamilton-Jacobi-Bellman (HJB)
> equation is a classical result in stochastic optimal control, where $s$
> represents the value function satisfying the HJB PDE by dynamic
> programming principles. We provide a detailed proof tailored to our setting in Section 6 of the
> paper. Specifically, the answer to your question is contained within
> lines 245--251. To improve clarity, we will explicitly introduce the HJB equation
> earlier in the manuscript and include standard references such as:
> *Bardi, M., and Capuzzo-Dolcetta, I. (1997). Optimal Control and
> Viscosity Solutions of Hamilton-Jacobi-Bellman Equations*.
>
> **Q3** : *What are EMA parameters, and target models in Section 4?*
>
> **Answer**
>
> EMA (Exponential Moving Average) parameters and target models are common
> techniques used in Reinforcement Learning (RL) (e.g., Deep Q-Learning)
> to promote training stability.
>
> In our setting, we aim to enforce the condition
> $$-\partial_t s(t, x) = -\frac{1}{2}\|\nabla_x s(t, x) \|^2 + U(x) + \frac{\sigma^2}{2}\text{tr} \{ \nabla^2 s(t, x) \}.$$
>
> A natural objective is then to minimize the squared residual:
> $$\min_s \left\|
>     \partial_t s(t, x) - \frac{1}{2}\|\nabla_x s(t, x)\|^2 + U(x) + \frac{\sigma^2}{2}\text{tr}\{\nabla^2 s(t, x)\}
> \right \|_2^2 .$$
>
> However, directly optimizing this objective tends to be unstable. To mitigate instability, we adopt a target model $\bar{s}$, whose parameters $\theta_{\text{target}}$ are updated as an exponential moving average of the online model parameters $\theta_{\text{online}}$:
> $$\theta_{\text{target}}
>     \leftarrow
>     \theta_{\text{target}} \cdot \alpha +
>     \theta_{\text{online}} \cdot (1 - \alpha),$$
> where $\alpha \in [0, 1)$ is the EMA decay rate.
> Using this target model, we reformulate the regression objective as a
> sum of two terms: $$\begin{aligned}
> \left\|
>     \partial_t \bar{s}(t, x) -\frac{1}{2}\|\nabla_x s(t, x) \|^2 + U(x) + \frac{\sigma^2}{2}\text{tr} \{ \nabla^2 s(t, x) \}
> \right\|_2^2 +  \left\|
>     \partial_t s(t, x) -\frac{1}{2}\|\nabla_x \bar{s}(t, x) \|^2 + U(x) + \frac{\sigma^2}{2}\text{tr} \{ \nabla^2 \bar{s}(t, x) \}
> \right\|_2^2
> \rightarrow \min_s.
> \end{aligned}$$ This formulation significantly stabilizes the training
> procedure.
>
> |       | Feasibility  |         |         | Optimality |         |         |
> |-------|----------------------------------------|---------|---------|----------------------------|---------|---------|
> |       | Stunnel                                 | Vneck   | GMM     | Stunnel                    | Vneck   | GMM     |
> | HOTA  | **0.006**                              | **0.002** | **0.19** | **320.90**                 | 115.09  | **80.44** |
> | HOTA w/o EMA | 0.018                              | 0.004   | 0.65    | 338.67                     | **109.25** | 97.30   |
>
> **Q4** : *The experiments focus mainly on toy datasets.
> Would it be possible to also do experiments on real data such as images?
> Are there real case scenario where there would be an interest to use non
> differentiable potential?*
>
> **Answer**
>
> Regarding real-case scenario with non differentiable potentials. One
> example is metric learning, mentioned in one of your questions.
> Specifically, if information about the metric is provided only as
> external environment feedback (so the potential function is not
> explicitly given), one can treat this feedback as a potential energy
> $U(x)$ and construct an optimal transport framework with cost
> $\int \left( \|v(t, x)\|_2^2 + U(x) \right) dt,$ and then perform metric
> learning by minimizing $\int \langle v(t, x), A(x) v(t, x)\rangle dt$
> with respect to $A$.
>
> Another relevant scenario arises when the potential is defined by
> non-differentiable with respect to tokens LLMs or a large neural network
> that propagating gradients through is computationally expensive. Our
> method requires only the evaluation of the potential outputs themselves,
> without backpropagating through the potential function, which may lead
> to a more efficient and scalable approach in such cases.
>
> Regarding benchmarks, we demonstrate HOTA's superior performance on
> standard 2-dimensional datasets and  1000-dimensional *opinion
> depolarization task* (presented in the supplementary materials), compared
> to previous state-of-the-art methods. We also evaluate our method on
> Sphere datasets across various dimensions.
>
> We attempted to apply HOTA to image translation tasks but encountered
> difficulties with stable convergence. As mentioned in the paper, the
> learned neural network $s(t, x)$ simultaneously represents two entities:
> the Kantorovich potential $s(1, x)$ and the control $-\nabla_x s(t, x)$.
> These two components are updated via distinct objectives, which modify
> the network parameters differently. This interplay can destabilize
> training, because improving the model with respect to one objective may
> degrade its performance concerning the other. This issue does not arise
> in classical one-step Optimal Transport solvers, where Kantorovich
> potential and transport maps are represented by separate neural
> networks.
>
> We partially addressed this issue by introducing adaptive gradient
> scaling, which improved stability to some extent. However, this approach
> alone is insufficient when dealing with more complex datasets such as
> images. A more thorough analysis of the interplay between the
> Kantorovich potential and control terms, as well as novel stabilization
> techniques, is needed to overcome these challenges. Investigating and
> developing solutions to this problem represents a promising direction
> for future research.
>
> You may ask: why use HOTA if methods like GSBM have shown success on
> image translation tasks?
>
> The key difference is that GSBM imposes strict limitations on the
> potential energy function: it must be differentiable and explicitly
> evaluable. Moreover, GSBM assumes Gaussian posterior distributions
> $p(x_t \mid x_0, x_1)$, which theoretically perform well mainly for
> quadratic potentials. To illustrate these limitations, we evaluated GSBM
> on our Sphere dataset and observed poor performance---while it reaches
> the target distribution, it fails to effectively circumvent the unit
> sphere potential.
>
> | Data dim         | 10   | 100  | 500  | 1000  |
> |------------------|------|------|------|-------|
> | HOTA L2UV        | **0.189**| **0.034**| **0.059**| **0.109** |
> | HOTA Optimality  | **3.37** | **3.16** | **3.78** | **3.17**  |
> | GSBM L2UV        | 0.31| 0.136 |  0.110  |  0.142   |
> | GSBM Optimality  | 4.04  | 6.27  | 20.88 | 22.1  |
>
> In conclusion, although HOTA currently struggles with image-based tasks,
> it supports working with significantly more complex and
> non-differentiable potentials while providing theoretical optimality
> guarantees.
>
> **Q5** : *Since the method can be used with general costs, it could also be a
> nice addition to perform metric learning as in \[1,2,3\].*
>
> **Answer**
>
> Metric learning and inverse optimal transport are indeed important and
> interesting tasks. However, the problem formulations addressed in the
> cited works differ from the setting for which HOTA was specifically
> designed.
>
> For example, in \[1, 2\], the metric is implicitly defined by samples of
> marginal distributions observed at multiple intermediate timestamps. The
> optimization alternates between solving dynamic OT problems locally
> between adjacent marginals with a fixed metric and updating the metric
> by minimizing the integral cost over the resulting dynamic optimal
> transports. Although our method could, in principle, be incorporated
> into such a pipeline, this setup does not reflect the primary design
> goals of HOTA. HOTA is a good at finding optimal transport paths
> autonomously by relying solely on system potential energy feedback from
> the environment, without requiring an explicit functional form of the
> potential or metric. Therefore, when the metric information is
> implicitly provided via environmental feedback, our method is
> well-suited and potentially optimal. However, this differs from the
> problem setting in \[1, 2\], rendering direct comparisons or
> integrations non-trivial and leaving this as a promising direction for
> future work.

---

> > ### Comment · Reviewer_N62o · 2025-08-04
> >
> > Than you for answering my questions. I now better understand the contributions of the paper and why it is not tested on images. I would recommend to clarify some of the points which where unclear such the background on optimal transport and HJB or EMA which might not be known to everyone interested into this paper. I would also suggest to add the limitations on the image tasks, and a paragraph on the possible applications for non differentiable potentials. I believe that one of the applications you mentioned would really strengthen this work.

---

> > > ### Author Response · Authors · 2025-08-04
> > >
> > > We are glad we could clarify  on these concerns. We commit to add the mentioned paragraph and the limitation for the images in the revised text. Thank you for your input.

---

### Official Review · Reviewer_Xycq · 2025-06-26

**Clarity:** 3
**Significance:** 2
**Originality:** 2
**Rating:** 4
**Confidence:** 2

**Summary:**

This paper propose a method to solve the Generalized Schrödinger Bridge (GSB) problem. Given a source distribution $\alpha$, a target distribution $\beta$, and a prior process, the Schrödinger Bridge (SB) problem seek the optimal dirft $v_t(x)$ that takes $\alpha$ to $\beta$, where in the dynamical formulation the cost is $||v_t(x)||^2$. GSB also allows potential $U_t(x)$ term in the cost. In contrast to [1], this work do not make any restricting assumption on the potential or on the marginal of the solution conditioned the edges (i.e., t=0 and t=1). This allows them find the optimal solution for larger class of problems but comes in price of higher computation cost. The paper present the dual formulation of the GSB in Theorem 1 and propose to optimize cost Eq. (8) by alleviating the constraint to soft constraint Eq. (9).  Algorithm for the training is presented and the method is validated on a number of small scale problems.

[1] Liu, Guan-Horng, et al. "Generalized Schr\" odinger Bridge Matching." arXiv preprint arXiv:2310.02233 (2023).

**Questions:**

1. Is the N-dimensional experiments  has same target and source distribution and just requires to transport the mean of the distribution between the poles? Additionally, why no other baseline is reported for this experiment?

2. Could you please provide a discussion/evaluation of computational cost or training times of your method vs [1]?

[1] Liu, Guan-Horng, et al. "Generalized Schr\" odinger Bridge Matching." arXiv preprint arXiv:2310.02233 (2023).

**Ethical Concerns:**

["NO or VERY MINOR ethics concerns only"]

**Final Justification:**

As i mentioned in my review the theoretical contribution of this paper is not fully clear to me since no clear separation between current work and previous work is provided.  Additionally, it seems the reported estimation of W2UV metric is not particularly meaningful in high dimension. Therefore, I remain unconvinced that the high-dimensional experiments clearly demonstrate the efficacy of the proposed method.

That said, in the paper and during the rebuttal period the authors have shown that their method consistently outperforms the main baseline, GSBM, across all metrics and in terms of training efficiency. Given that the generalized Schrödinger bridge (GSB) problem lies somewhat outside my main area of expertise (as I noted at the beginning of my review), I consider outperforming the best available baseline a sufficient justification for a positive rating.

**Limitations:**

yes

**Paper Formatting Concerns:**

.

**Quality:**

2

**Strengths And Weaknesses:**

I will begin by noting that I was not eligible for biding on papers as a reviewer. Although the topic of the paper lies outside my primary research domain, I have made a sincere effort to evaluate it fairly and thoughtfully.

> Strengths:
1. The general formulation of the proposed method allows to solve a large class of problems.

> Weaknesses:
1. Though no reference was given, Theorem 1 is provided in the preliminaries section and hence I assume it was proved in previous works. Thus the main contribution of the paper is simply writing the estimator of the terms in the theorem and alleviating the constraint to a soft constraint. I find this contribution of low novelty, since no additional lemmas or theorems were need to write the provided estimator.

2. It seems the provided estimator is not scalable:
 - It requires simulation of the process during training. The author is proposing to use a buffer which potentially would solve this problem, but it is not clear if this is the case and no discussion on the matter is done by the authors.
- The network parametrize the value function $s(x,t)$, hence the optimization will require second order derivative which tend to be untractable in larger scale.
- The problem of the high dimension seems relatively easy, i.e., only transporting the mean of the distribution on the sphere.

---

> ### Author Rebuttal · Authors · 2025-07-31
>
> **Weaknesses**
>
> **W1** : *Though no reference was given, Theorem 1 is provided in the
> preliminaries section and hence I assume it was proved in previous
> works. Thus the main contribution of the paper is simply writing the
> estimator of the terms in the theorem and alleviating the constraint to
> a soft constraint. I find this contribution of low novelty, since no
> additional lemmas or theorems were need to write the provided
> estimator.*
>
> **Answer**
>
> Thank you for this important remark. We appreciate the opportunity to
> clarify the presentation and contributions related to Theorem 1.
>
> While Theorem 1 currently appears in the  "Preliminaries" section
> of the paper, we want to emphasize that both the theorem and its proof
> are novel contributions of our work. To the best of our knowledge, this
> specific dual formulation of the generalized Schrödinger bridge problem,
> which tightly connects the Kantorovich potential with the
> Hamilton-Jacobi-Bellman PDE framework as presented in our Theorem 1, has
> never been previously reported in the literature.
>
> The  "Preliminaries" section was intended to introduce necessary
> background and a formal framework, along with a brief explanation of the
> core idea behind our method. However, we acknowledge that this placement
> may lead to confusion regarding the originality of the theoretical
> results. Relocating the theorem and this explanation to a dedicated
> "Theoretical Grounding" subsection within the  "Method"
> section would better highlight its novelty and clarify its role.
>
> We will revise the manuscript accordingly to ensure this important
> contribution is clearly emphasized and properly referenced as original
> work.
>
> **W2** : *It seems the provided estimator is not scalable:*
>
> -   *It requires simulation of the process during training. The author
>     is proposing to use a buffer which potentially would solve this
>     problem, but it is not clear if this is the case and no discussion
>     on the matter is done by the authors.*
>
> -   *The network $s(t, x)$ parametrize the value function , hence the
>     optimization will require second order derivative which tend to be
>     untractable in larger scale.*
>
> -   *The problem of the high dimension seems relatively easy, i.e., only
>     transporting the mean of the distribution on the sphere.*
>
> **Answer**
>
> Thank you for giving us the opportunity to defend scalability of our
> method.
>
> In actuality, the simulations overhead is minimal in practice, scaling
> linearly with the number of steps (see table below). The replay buffer
> reduces the required simulations by reusing past trajectories, such that we train the HJB loss on the buffer data and use simulations only for the potential loss. This results in speed improvement of approximately 1.5x.
> However, the primary speedup comes from JAX's scan-based optimization of
> the simulation loops.
>
> Remarkably, JAX's
> automatic differentiation computes $\nabla s$ and $\Delta s$ in
> **near-identical time** to $s(t,x)$ itself (\<5% overhead), verified
> across dimensions ($10^2, 10^4, 10^5$) using a 4-layer
> MLP$(512, 512, 512, 1)$.
>
> Additionally, to further illustrate efficiency, we conducted experiments
> on our Sphere dataset. We fixed the number of training iterations at
> 70,000 and varied the data dimensionality (with simulation steps fixed
> at 30) as well as varied the number of simulation steps (with data
> dimensionality fixed at 1000). The results in the tables below show that
> HOTA scales well (near-linear) with both increasing dimensionality and
> increasing number of simulation steps.
>
>  | Data dim       |    1k   |    3k   |    5k   |    7k    |    9k    |
> |:---------------|--------:|--------:|--------:|---------:|---------:|
> | HOTA time      | 4.4 min | 7.1 min | 9.9 min | 12.8 min | 15.7 min |
>
> | Simulation steps |   10    |   20    |   30    |   40     |   50     |
> |:-----------------|--------:|--------:|--------:|---------:|---------:|
> | HOTA time        | 3.4 min | 3.7 min | 4.4 min | 5.8 min  | 7.1 min  |
>
> We hope this comprehensive analysis addresses your question about the
> computational cost and scalability of our method. We propose to enhance the Experiments
> Section  of our paper with these data.
>
> Besides that, it is possible to make a
> simulation-free extension of our method via integral drift distillation,
> which we will explore in future work:
> $$u(x, t_1, t_2) = \int_{t_1}^{t_2} v(t,x) \, dt$$
>
> **Questions**
>
>  **Q1** : *Is the N-dimensional experiments has same target and source
> distribution and just requires to transport the mean of the distribution
> between the poles? Additionally, why no other baseline is reported for
> this experiment?*
>
> **Answer**
>
> Thank you for your question regarding the Sphere dataset.
>
> Let us clarify the problem setup. System potential is defined as open
> sphere with radius 1: $$\begin{aligned}
>     U(x) =  C_d \mathbf{I} (\|x\|_2 < 1), \text{ where } C_d \text{ is a dimension-dependent constant.}
> \end{aligned}$$ The source and target distributions lie at the opposite
> poles of the sphere, $p_1 = (1, 0, \ldots, 0)$ and
> $p_2 = (-1, 0, \ldots, 0)$, respectively. Specifically, to obtain the
> source distribution, we first sample from a normal distribution
> $x \sim \mathcal{N}(p_1, \sigma)$ and then project onto the unit sphere
> by normalizing: $\frac{x}{\|x\|}$. The same we do to get target sample.
> We also consider a smooth variant of $U(x)$ to test sensitivity to the smoothness parameter and in comparison with GSBM baseline (see table below). The  smooth  $U(x)$ is defined as
> $$C_d \text{sigmoid}((1 - |x|_2) / \tau ) $$
>
> The objective is to find a transport dynamics that moves the source
> distribution to the target distribution while circumventing the sphere
> potential. The Sphere dataset presents a challenging problem, especially
> because the optimal paths do not overlap with simple interpolations
> between the distributions that require additional exploration.
>
> Regarding the absence of additional baseline comparisons on this
> high-dimensional dataset, our primary focus was to demonstrate HOTA's
> scalability and flexibility across varying dimensionalities, having
> already established its superiority on other benchmark problems.
>
> Nevertheless, we evaluated our main rival, GSBM, on this Sphere task.
> The results (see table below) indicate that while GSBM can approximately
> match the target distribution, it struggles to find an optimal path
> respecting the spherical barrier. This may be explained by GSBM's
> difficulties in learning conditional intermediate distributions
> $p(x_t \mid x_0, x_1)$, which break down when initial trajectories miss
> optimal transport regions.
>
> | Data dim         | 10   | 100  | 500  | 1000  |
> |------------------|------|------|------|-------|
> | HOTA L2UV        | **0.189**| **0.034**| **0.059**| **0.109** |
> | HOTA Optimality  | **3.37** | **3.16** | **3.78** | **3.17**  |
> | GSBM L2UV        | 0.31| 0.136 |  0.110  |  0.142   |
> | GSBM Optimality  | 4.04  | 6.27  | 20.88 | 22.1  |
>
> **Q2** : *Could you please provide a discussion/evaluation of computational
> cost or training times of your method vs \[1\]?*
>
> **Answer**
>
> Thank you for your question regarding the computational cost and
> training times of our method compared to GSBM. We understand your
> concerns about model scalability; however, our empirical results show
> that these concerns do not hold in practice. Here is the requested
> direct comparison of training times between our method (HOTA) and the
> baseline GSBM method. As shown in the table below, HOTA is more
> efficient in both low-dimensional (2D) and high-dimensional (1000D)
> problems.
>
> | Datasets                  | 2D benchmark | 1000D opinion depolarization |
> |---------------------------|--------------|-------------------------------|
> | GSBM time                 | 18-45 min    | 870 min                     |
> | HOTA (ours) time          | 6-7 min      | 26 min                        |

---

> > ### Comment · Reviewer_Xycq · 2025-08-04
> >
> > First, I thanks the authors for their response.
> >
> > Regarding, the hyper-sphere experiment, a single Gaussian distribution projected to a hyper-sphere still seems like a very naive setup. Why not considering a more complex distribution?
> >
> > Regarding the training times, can you please further detail on where is the advantage of your method is coming from and not just the numbers?

---

> > > ### Author Response · Authors · 2025-08-04
> > >
> > > Thank you for your response!
> > >
> > > We employ a relatively simple distribution in our experiments because the primary focus is on analyzing learning robustness under increasing dimensionality in the Spheres environment. Prior HJB-based methods have shown significant degradation in performance with higher dimensions. To explicitly demonstrate our method's effectiveness in such settings, we designed this problem family to provide clear and scalable empirical validation. In general case it is not possible to define the reference mapping $T^*(x)$ used in L2UV metric.
> > >
> > > DeepGSB, one of the main representatives of HJB-based approaches currently, fails to solve the Opinion Depolarization (1000D) task, while our approach achieves state-of-the-art results:
> > >
> > > | Method                  | Optimality |
> > > |-------------------------|------------|
> > > | DeepGSBM (HJB-based)    | 914.2      |
> > > | HOTA (ours, HJB-based)  | **460.4**  |
> > > | GSBM                    | 505.3      |
> > >
> > > Concerning the perceived simplicity of the Sphere task, as mentioned, the GSBM method was unable to solve it successfully (see the table below). Thus, the assumption that the problem is simple may not hold in practice.
> > >
> > > | Data dim         | 10   | 100  | 500  | 1000  |
> > > |------------------|------|------|------|-------|
> > > | HOTA L2UV        | **0.189**| **0.034**| **0.059**| **0.109** |
> > > | HOTA Optimality  | **3.37** | **3.16** | **3.78** | **3.17**  |
> > > | GSBM L2UV        | 0.31| 0.136 |  0.110  |  0.142   |
> > > | GSBM Optimality  | 4.04  | 6.27  | 20.88 | 22.1  |
> > >
> > > Please let us know if you’d be interested in conducting an additional experiment involving spheres with a more complex distribution. While this will require some time, we can complete it before the end of the discussion.
> > >
> > > **Regarding computational efficiency**, our method achieves a good training speed through the aforementioned technical implementations (including JAX-optimized loops) combined with linear scaling with input dimensionality.
> > >
> > > Compared to GSBM, our method achieves superior computational efficiency by learning a single entity--the value function $s(t,x)$, while GSBM alternates between control optimization and density estimation, incurring additional computational overhead. It is well-established that density estimation becomes significantly more challenging as the dimensionality of the space increases.
> > > In certain experiments, GSBM additionally employs the congestion loss that scales quadratically with the batch size, further increasing its computational overhead.
> > >
> > > We have aggregated the main differences between these methods (including those affecting the running time) in the following table.
> > >
> > > | Property               | HOTA                   | GSBM                  |
> > > |------------------------|------------------------|-----------------------|
> > > |Training entities       | Value function   $s(t, x)$      |      Density $\rho(t,x)$, flow-matching $u(x)$        |
> > > | Optimality criterion |          Integral         |   Element-wise    |
> > > | Target matching criterion | Kantorovich potentials |   Heuristic    |
> > > | Additional costs | Acceleration |   Entropy, Congestion   |
> > > | Alternating optimization        | No                     | Yes      |
> > > | Simulation-free                 | No, but with minor overhead                         |  Yes     |

---

> > > ### Author Response · Authors · 2025-08-06
> > >
> > > As requested, we have created an additional dataset with more complex distribution to better evaluate the robustness and dimensional scalability of our method. The new dataset consists of a mixture of Gaussian distributions on a high-dimensional sphere. As previously, the data is distributed around two antipodal regions (north/south poles) with K=10 clusters per pole.  Cluster centers are perturbed randomly from the poles using a controllable spread parameter.  Points are sampled by adding a Gaussian noise to cluster centers.
> > >
> > > We evaluate our method (HOTA) against the GSBM baseline across increasing dimensionalities, measuring two key metrics:
> > >
> > > W2UV: Wasserstein-2 distance between predicted and target distributions normalized by variance of the target;
> > > Optimality: Lower values reflect more efficient transport plans.
> > >
> > > | Data dim         | 10   | 100  | 500  | 1000  |
> > > |------------------|------|------|------|-------|
> > > | HOTA W2UV        | **0.02**| **0.81**| 1.33 | 1.40 |
> > > | HOTA Optimality  | **2.40** | **2.76** | **3.24** | **3.09**  |
> > > | GSBM W2UV        |   0.25  |    0.87     |   **0.91**     |  **0.95**   |
> > > | GSBM Optimality  |    5.31   |   6.17   |   14.59   |   14.63    |
> > >
> > > We would be grateful for the opportunity to further discuss this new dataset and how it might enhance our analysis.

---

> > > > ### Comment · Reviewer_Xycq · 2025-08-06
> > > >
> > > > I want to thank the authors for providing a more detailed analysis of the method's computational efficiency. Indeed they have shown that HOTA enjoys better computational efficiency than GSBM.
> > > >
> > > > Regarding the new dataset, to my understanding GSBM does claim to produce the optimal transport plan hence it is not surprising that HOTA results in better optimality.  Is there no other baseline to compare with that tried to learn the optimal plan?

---

> ### Author Response · Authors · 2025-08-05
>
> We present further rigorous evaluation of the computational advantage of our method. Below, we conduct a direct runtime comparison on the Spheres dataset with varying data dimensions from 1k to 9k.
>
> HOTA achieves significant speedups over GSBM due to (1) a simpler optimization objective and (2) better dimensional scalability. Speedup factors (rightmost column) are calculated as the ratio of GSBM’s runtime to HOTA’s runtime.  Notably, GSBM also encounters out-of-memory (OOM) errors beyond 5,000 dimensions under 24GB GPU memory constraints, while HOTA maintains stable performance through 9,000 dimensions.
>
> | Dimension | HOTA Time (min) | GSBM Time (min) | Speedup (×) |
> |-----------|-----------------|-----------------|-------------|
> | 1,000     | 4.4             | 241             | 54.8        |
> | 3,000     | 7.1             | 620             | 87.3        |
> | 5,000     | 9.9             | 1233            | 124.5       |
> | 7,000     | 12.8            | OOM             | -           |
> | 9,000     | 15.7            | OOM             | -           |

---

> ### Author Response · Authors · 2025-08-06
> **GSB Baselines Clarifications**
>
> Thank you for acknowledging our clarifications. With regard to the lack of established baselines, you raise a very important point. Indeed, neural-based approaches for solving Generalized Schrödinger Bridge (GSB) problems represent an emerging yet rapidly developing research area. Currently, GSBM stands as a clearly acknowledged state-of-the-art method, particularly in optimal transport plan learning, as evidenced by both the original GSBM publication [1] and our comparative experiments.
>
> In our evaluation, we have compared HOTA against other relevant baselines including:
>
> 1. NLSB [2] & NLOT [3]: While pioneering in solving GSB problems, these methods exhibit significant limitations: unstable performance across different scenarios, poor scalability beyond low-dimensional spaces, divergence when handling complex potential functions $U(x)$.
>
> 2. WLF [4]: Though theoretically similar to GSBM, practical implementations have shown suboptimal empirical performance and convergence issues in high-dimensional settings. We assume that this is because of requirement on differentiability of potential function.
>
> 3. DeepGSB [5]: This HJB-based approach has been rigorously compared against GSBM in prior work. We have included their published results on depolarization task into our experimental evaluation, but faced technical problems when adapting their code to our new datasets.
>
> 4. We also may compare to ENOT [6], a static OT SOTA model from the last year’s spotlights on the NeurIPS. It solves a simpler problem, but with its help we can obtain reference values of optimality, knowing the metric on the Sphere. If you’d like we can include this result as well, even though the static problem formulation is somewhat beyond the scope of our study.
>
> Given we overperform GSBM on the both HD Spheres datasets, we are absolutely positive the other models will lag behind even more. Still, we could include the values in the revised text given the opportunity.
>
> [1] Yaron Lipman, Ricky T. Q. Chen, Heli Ben-Hamu, Maximilian Nickel, and Matthew Le. 2023 "Flow matching for generative modeling".
>
> [2] Koshizuka and Sato, 2022 "Neural Lagrangian Schrödinger Bridge: Diffusion Modeling for Population Dynamics".
>
> [3] Aram-Alexandre Pooladian, Carles Domingo-Enrich, Ricky T. Q. Chen, and Brandon Amos. 2022 "Neural optimal transport with Lagrangian costs".
>
> [4] Kirill Neklyudov, Rob Brekelmans, Alexander Tong, Lazar Atanackovic, Qiang Liu, and Alireza Makhzani. 2024 "A computational framework for solving Wasserstein Lagrangian flows".
>
> [5] Guan-Horng Liu, Tianrong Chen, Oswin So, and Evangelos A Theodorou. 2022 "Deep generalized schrödinger bridge".
>
> [6] Nazar Buzun, Maksim Bobrin, Dmitry V. Dylov,  2025 "ENOT: Expectile Regularization for Fast and Accurate Training of Neural Optimal Transport".

---

> > ### Comment · Reviewer_Xycq · 2025-08-06
> >
> > If the authors could clarify why they claim that HOTA outperform GSBM in the new experiments on the sphere? It appears from the table that at larger dimensions the GSBM achieves a better Wasserstein-2 distance.

---

> > > ### Author Response · Authors · 2025-08-06
> > >
> > > 1) We emphasize Optimality scores in our analysis because Wasserstein-2 distance becomes less informative in high-dimensional spaces. Unlike our first experiment, this spherical dataset lacks reference point-to-point mappings (ground truth $x_0 \to x_1$ correspondences), making L2-based metrics impossible to compute. Optimality provides more meaningful comparison as it directly evaluates transport plan cost.
> > > 2) While GSBM achieves marginally better W2 distances at 500D/1000D, this comes with significantly worse path optimality. We hypothesize this occurs because GSBM likely violates spherical constraints to minimize W2, creating geometrically inefficient paths. Besides that, HOTA maintains stable optimality (~3.0 across all dimensions).
> > >
> > > This would have been much easier to show on actual transport maps, which unfortunately we are not allowed to attach here.

---

> > > > ### Comment · Reviewer_Xycq · 2025-08-06
> > > >
> > > > 1. The claim "Optimality provides more meaningful comparison as it directly evaluates transport plan cost." is not exact since the transport plan cost is also dependent on the resulted target distribution and its clear that the plan optimality can be improved on the cost distribution distance.
> > > >
> > > > 2. Could the authors please provide an explanation on how significant is the W2 difference between 0.91 vs 1.33 and 0.95 vs 1.40? For example what is the W2 difference of the target distribution and the source or a uniform distribution?

---

> ### Author Response · Authors · 2025-08-07
> **HOTA outperforms GSBM in the new experiments on the sphere**
>
> We fully agree that transport plan optimality depends on the resulting distribution of $x_1$. Our claim that "Optimality provides a more meaningful comparison" is made under the condition that the distribution of $x_1$ obtained from HOTA's solution closely matches the target distribution. This assumption is supported by:
>
> 1) Theoretical guarantees (Theorem 1) proving convergence to the target distribution when we optimize the Kantorovich potentials.
>
> 2) Empirical validation across all experiments presented in out paper.
>
> To better understand the behavior of the W2UV metric, we conducted a systematic evaluation by computing self-consistency values W2UV(tgt, tgt) between two independent samples (size $N=10^4$) from the target distribution $\beta$ and W2UV(src, tgt) between samples from source and target distributions.
>
> | Dimension | W2UV(tgt, tgt) | W2UV(src, tgt) |
> |----------------|-------------|-------------|
> | 10             | 0.005       | 5.07        |
> | 100            | 0.63        | 1.73        |
> | 500            | 1.29        | 1.71        |
> | 1000           | 1.37        | 1.75        |
>
> This analysis confirms that while W2UV remains a useful metric, its utility diminishes in high-dimensional spaces, motivating our focus on the Optimality metrics.  Nevertheless, the HOTA values at 500D/1000D (1.33 and 1.40 respectively) approach the reference lower bounds (1.29 and 1.37).
>
> We found that the definition of the W2 metric from the implementation of the GSBM algorithm contains an additional multiplier of 0.5, so that its why their metric values could be smaller than the lower bound. If we correct for this factor in the GSBM algorithm for an exact apple-to-apple comparison, the table will look like:
>
> | Data dim         | 10   | 100  | 500  | 1000  |
> |------------------|------|------|------|-------|
> | HOTA W2UV        | **0.02**| **0.81**| **1.33** | **1.40** |
> | HOTA Optimality  | **2.40** | **2.76** | **3.24** | **3.09**  |
> | GSBM W2UV        |   0.50  |    1.74     |   1.82     |  1.90   |
> | GSBM Optimality  |    5.31   |   6.17   |   14.59   |   14.63    |
>
> Please note that our claims on applicability of the optimally measure and the overall dynamics as a function of dimensionality remain intact. And after this correction we can definitely say that HOTA outperform GSBM in all new experiments on the sphere.

---

> > ### Comment · Reviewer_Xycq · 2025-08-08
> >
> > I would like to thank the authors for the additional results and detailed explanations provided during the rebuttal and discussion period.
> >
> > It appears that the estimated W2UV metric is not particularly meaningful in high dimensions, and in fact, the W2UV score of the GSBM method seems worse than simply using the source distribution (which will results in perfectly optima transport cost, i.e., not moving at all..). Therefore, I remain unconvinced that the high-dimensional experiments clearly demonstrate the efficacy of the proposed method.
> >
> > That said, the authors have shown that their method consistently outperforms the main baseline, GSBM, across all metrics and in terms of training efficiency. Given that the generalized Schrödinger bridge (GSB) problem lies somewhat outside my main area of expertise (as I noted at the beginning of my review), I consider outperforming the best available baseline a sufficient justification for a positive rating, and I will revise my rating accordingly.

---

> > > ### Author Response · Authors · 2025-08-09
> > >
> > > We sincerely thank the reviewer for the careful attention to our work. We are pleased that we were able to answer key questions regarding theoretical contribution, computational efficiency, and comparison with GSBM.
> > >
> > > To strengthen our high-dimensional results on the last sphere experiment and demonstrate the alignment of the target distribution we will include 2D projections and directional similarities histograms of the transported distributions (similar to our analysis of the 1000D depolarization problem, Fig. 1 in the supplementary material).

---

### Official Review · Reviewer_Tp52 · 2025-07-03

**Clarity:** 4
**Significance:** 3
**Originality:** 3
**Rating:** 5
**Confidence:** 3

**Summary:**

The authors formulate and prove equivalence of a dual formulation of the generalized Schrodinger bridge problem. This formulation allows one to avoid explicit representation of the evolving densities and to generate trajectories directly with an Euler-Maruyama approach. Also, it can handle non-smooth potentials and non-differentiable cost functions.

They then design a method for solution via network parameterization of the value function and an RL approach. Their loss function combines potential terms with a Hamiltonian loss. The network is an MLP with Fourier feature encoding of the time component (with a frequency cap). Empirically, they show that their method outperforms existing methods in terms of matching the target distribution and path optimality.

**Questions:**

1. Is there any potential for getting stuck in local minima by the initiation of the approximate flow region as linear interpolation? I.e., is it possible for the trajectories to be attracted to a higher cost path that happens to be closer to this initialization, rather than finding a more distant lower cost path?
2. Can you be more specific on the potential and smoothness parameter for the Sphere dataset? This seemed a bit vague unless I missed something.

**Ethical Concerns:**

["NO or VERY MINOR ethics concerns only"]

**Limitations:**

Yes.

**Paper Formatting Concerns:**

None.

**Quality:**

3

**Strengths And Weaknesses:**

Strengths:
1. The problem at hand is an important one and they come up with a formulation that handles non-smooth potentials and non-differentiable cost functions.
2. The method outperforms the SOTA on some benchmark datasets.
3. Pseudocode is clearly provided as well as experimental details, so reproduction of their method should be achievable.

Weaknesses:
1. It might be nice to have a comparison with competing methods on computational cost / training time and the rate of convergence of the results.
2. Perhaps, the method could be tested and compared on more datasets. There is only a single high-dimensional example, and I could envision more challenging examples in the planar setting (perhaps a not so "baby" maze).

---

> ### Author Rebuttal · Authors · 2025-07-31
>
> We are thankful for the positive feedback and for raising important questions. We address them point to point below.
>
> **Comparison with competing methods on computational cost / training
> time and the rate of convergence of the results.**
>
> We compared the training time with the main baseline GSBM on the same
> hardware. Due to the limitations of this conference we cannot present
> the graphical results with convergence rates in the Rebuttal but will
> add them to the article.
>
> | Datasets                  | 2D benchmark | 1000D opinion depolarization |
> |---------------------------|--------------|-------------------------------|
> | GSBM time                 | 18-45 min    | 870 min                     |
> | HOTA (ours) time          | 6-7 min      | 26 min                        |
>
> **The method could be tested and compared on more datasets.**
>
> In the supplementary material we provide additional experiments on
> **opinion depolarization task** in high-dimensional setting ($R^{1000}$) and
> compare with baselines DeepGSB Liu et al. \[2022\] and GSBM Liu et al.
> \[2024\].
>
> Additionally, we made  comparisons with the main baseline GSBM on high-dimensional spheres with potential $U_\text{smooth}(x)$ and $\tau=0.01$ (smoothness coefficient).  We used the Optimality and L2UV metric (from A. Korotin et. al "Do Neural Optimal Transport Solvers Work?".) instead of Feasibility.
>
> | Data dim         | 10   | 100  | 500  | 1000  |
> |------------------|------|------|------|-------|
> | HOTA L2UV        | **0.189**| **0.034**| **0.059**| **0.109** |
> | HOTA Optimality  | **3.37** | **3.16** | **3.78** | **3.17**  |
> | GSBM L2UV        | 0.31| 0.136 |  0.110  |  0.142   |
> | GSBM Optimality  | 4.04  | 6.27  | 20.88 | 22.1  |
>
> **Is there any potential for getting stuck in local minima by the
> initiation of the approximate flow region as linear interpolation?** Not
> really. Linear interpolation between the source and target distributions
> is used as an initial rough approximation of the flow region. This
> heuristic helps bootstrap training by providing a reasonable starting
> point for the replay buffer B. During hyperparameters tuning, we
> occasionally encountered such local minima in the training dynamics
> (generated trajectories pass through regions of high potential $U(x)$).
> However, in all cases, this issue was successfully resolved by
> increasing the weight of $U(x)$. Even if the initial trajectories from
> linear interpolation miss critical regions of the state space (like in
> Stunnel of Spheres dataset), the optimization produces new states beyond
> the initialization.
>
> From a theoretical point of view, the coercivity condition (eq. 11)
> ensures the Hamilton-Jacobi-Bellman (HJB) equation (eq. 9) admits a
> unique, well-behaved solution. This condition implicitly penalizes
> overshooting gradients and high curvature, which could arise from poor
> initialization (e.g., linear interpolation crossing a high-cost
> barrier).
>
> **Can you be more specific on the potential and smoothness parameter for
> the Sphere dataset?**
>
> Certainly. The sphere experiments evaluate the transport of
> distributions between antipodal (opposite) poles of hyperspheres in
> $\mathbb{R}^d$, where dimensionality $d$ scales as
> $d \in {3, 9, 27, \dots, 729}$. The source and target measures are
> Gaussian distributions projected onto the sphere's surface. To enforce
> geometric constraints, we introduce a potential $U(x)$ that penalizes
> deviations from the manifold. We study two variants:
> $$U_\text{sharp}(x) =  C_d \mathbf{I} (\|x\| < 1)$$ where $C_d > 0$ is a
> dimension-dependent constant, and
> $$U_\text{smooth}(x) =  C_d \, \text{sigmoid} ((1 - \|x\|) / \tau)$$ The
> sharp potential tests scalability under strict constraints, while the
> smooth variant evaluates sensitivity to the relaxation parameter $\tau$
> (Figure 2.b).
>
> We will include this elaboration in the revised text. Thank you.

---

### Official Review · Reviewer_yN65 · 2025-07-05

**Clarity:** 3
**Significance:** 2
**Originality:** 3
**Rating:** 3
**Confidence:** 3

**Summary:**

The authors propose a method for solving the Schrodinger Bridge problem with dynamical state costs by minimizing (i) the violation of the Hamiltion-Jacobi equation along a trajectory and (ii) relevant boundary terms.   The authors propose several techniques for stabilizing optimization dynamics, such as separating the time- and space- derivative regression targets using an EMA network and scaling the contribution of gradients arising from (i) and (ii).   These choices are ablated on several two-dimensional datasets.

**Questions:**

I would be interested in an ablation which removes the EMA splitting of the objective, with all terms involving $s_\theta$ optimized together.

I would also be interested in acceleration and replay buffer ablations for the BabyMaze dataset.   In principle, it seems odd to introduce (i) straight-path inductive bias and consider (ii) fixed linear interpolation sampling (buffer-free), since the main feature and challenge of problems with state-cost is that the straight paths are *not* the solution.     I'd like to see the authors discuss these questions in the context of Table 2 / Figure 3 ablations, perhaps recommending that an expectation of "reasonably straight paths" may encourage high $\lambda_a$, for example.





**Minor Comments**

Places where notation might be cleaned up?

- lines 89-92 ($\alpha = \delta_x$?   where does $\beta$ appear?)

- $\mu_{v}(\cdot | x)$ could be written as a conditional distribution / map induced by the SDE with drift $v$?    $\mathbb{E}_{y \sim \mu(x)}[c(x,\mu)]$ is confusing notation, and  $c(x,\mu)$ in Eq. 2 depends on the entire trajectory, not just the static map.

- Eq 6 $\inf$ over $x_t$ appears to be incorrect

- different notation should be chosen for the loss-gradient scaling term $\alpha$ vs. initial marginal distribution $\alpha$

What is NLSB as a baseline?    I thought it might be Koshizuka & Sato 2024, "Neural Lagrangian Schrödinger Bridge" which might merit citation.   Pooladian et. al 2024 solves the deterministic problem.


In Table 2, what does "gray values correspond to the method's divergence" mean?   It seems these (lower) values are not to be trusted?

**Ethical Concerns:**

["NO or VERY MINOR ethics concerns only"]

**Limitations:**

The authors discuss sensitivity to the Fourier feature encoding of time, and provide ablation results suggesting the necessity of design choices such as adaptive gradient scaling and replay buffers based on simulated trajectories.

However, compared to diffusion-based models or GSBM, which parameterize the drift function directly as a neural network, HJB-based methods require a scalar neural network $s(t,x) \in \mathbb{R}$ such that we can take derivatives $\partial_t s, \nabla_x s, \Delta_x s$, adding computational cost.

What parameterization was used for the scalar function?   This may also constitute an important design choice.   See discussion in Salimans & Ho 2021 "Should EBMs Model the Energy or the Score?", Neklyudov et. al 2023 "Action Matching" and 2024 "Wasserstein Lagrangian Flows", Thornton et. al 2025 "Composition and Control with Distilled Energy Diffusion Models and SMC".

I cannot find the theoretical guarantees claimed in line 48, although none immediately come to mind as feasible and relevant.

**Quality:**

3

**Strengths And Weaknesses:**

** Strengths **
The authors provide a nice analysis of the c-duality associated with Lagrangian OT costs.   Several clever tricks for optimization are presented, and it is interesting that the method works as is for non-differentiable costs.

** Weaknesses **
The distinction between unregularized OT and regularized OT is blurred in this work, where we invoke the unregularized dual potential perspective for optimizing over SDEs.    It seems this could be addressed by reparameterization $u_t = v_t - \frac{\sigma_t^2}{2} \nabla \log p_t$ (e.g. [1] Ex. 4.4) and adding an additional term into the cost $U(x)$ involving $p_t(x)$.   However, this also suggests the proposed method may be equally valid for deterministic transport maps.

Experiments are fairly limited, focused on small two-dimensional examples.   I did not fully understand the Spheres dataset example and more detail on the dataset would be appreciated.   Are there complications arising from working on a manifold?

Ultimately, I am unsure of the significance of this work due to limited experimental evaluation.

[1] Neklyudov et. al 2024 "Wasserstein Lagrangian Flows"

---

> ### Author Rebuttal · Authors · 2025-07-31
>
> **Weaknesses**
>
> **W1: Blurred Distinction Between Regularized and Unregularized OT**. We
> appreciate the insightful suggestion regarding the regularization of
> dual potentials and the value function $s(t,x)$. The proposed
> reparameterization from Neklyudov et. al 2024 paper indeed offers a
> valuable theoretical connection to deterministic transport maps. Below
> we present the complete derivation and discuss its implications.
> Consider the following reparameterized value function: $$\tag{1}
> V(t,x) = s(t,x) + \frac{\sigma^2}{2} \log \rho(t,x)$$ where $s(t,x)$
> solves the original HJB equation $$\tag{2}
> \partial_t s = \frac{1}{2} \|\nabla s\|^2 - U(x) - \frac{\sigma^2}{2} \Delta s$$
> The time derivative after the reparameterization is $$\tag{3}
> \partial_t V = \partial_t s + \frac{\sigma^2}{2} \partial_t \log \rho$$
> From Fokker-Planck (with drift $v = -\nabla s$) equation we obtain that
> $$\partial_t \rho = \nabla^T (\rho \nabla s) + \frac{\sigma^2}{2} \Delta \rho$$
> and for $\log \rho$
> $$\partial_t \log \rho = \Delta s + \nabla s \cdot \nabla \log \rho + \frac{\sigma^2}{2} (\Delta \log \rho + \|\nabla \log \rho\|^2)$$
> Substitute it into $\partial_t V$ (3). After cancellation of $\Delta s$
> terms from equations (2) and (3) we get
> $$\partial_t V = \frac{1}{2} \|\nabla s\|^2 - U(x) + \frac{\sigma^2}{2} \nabla s \cdot \nabla \log \rho + \frac{\sigma^4}{4} \Delta \log \rho + \frac{\sigma^4}{4} \|\nabla \log \rho\|^2$$
> Substitute into the last equation the following expressions derived from
> (1):
> $$\|\nabla s\|^2 = \|\nabla V\|^2 - \sigma^2 \nabla V \cdot \nabla \log \rho + \frac{\sigma^4}{4} \|\nabla \log \rho\|^2$$
> $$\nabla s \cdot \nabla \log \rho = \nabla V \cdot \nabla \log \rho - \frac{\sigma^2}{2} \|\nabla \log \rho\|^2$$
> Combine all terms and obtain HJB equation for $V(x, t)$: $$\boxed{
> \partial_t V = \frac{1}{2} \|\nabla V\|^2 - U(x) + \frac{\sigma^4}{8} \|\nabla \log \rho\|^2 + \frac{\sigma^4}{4} \Delta \log \rho
> }$$ It corresponds to the *deterministic* transport map produced by the
> drift $v(t, x) = -\nabla V(t, x)$, such that
> $$\partial_t \rho = \nabla^T (\rho \nabla V)$$ In the reparameterized
> HJB the Laplacian $\Delta s$ cancels out, but its effects are absorbed
> into terms $\|\nabla \log \rho\|^2$ (local density gradients) and
> $\Delta \log \rho$ (density curvature). This demonstrates that
> $\Delta s$ implicitly encodes how noise influences the density evolution
> over time. And that is the difference between regularized and
> unregularized OT.
>
>
> **W2: Experiments are fairly limited, focused on small two-dimensional
> examples.** Our focus on 2D benchmarks (e.g., Stunnel and Vneck from
> \[Liu et al., 2024\]) was intentional, as these datasets are widely
> adopted in the Generalized Schrödinger Bridge (GSB) literature for their
> ability to enable direct visualization of particle trajectories and
> facilitate rigorous comparison with prior work (e.g., \[Liu et al.,
> 2022, 2024\]), where these benchmarks serve as established baselines.
>
> *The high-dimensional experiments*, besides N-dimensional unit spheres,
> include frequently used opinion depolarization task (covered
> comprehensively in supplementary material). Here, our approach
> outperforms DeepGSB \[Liu et al., 2022\] and GSBM \[Liu et al., 2024\],
> particularly in preserving consensus dynamics (Figure 1, supplementary).
> Notably, the previous HJB-based approach DeepGSB struggles with
> polarization due to its suboptimal target matching criterion.
>
> As requested, we made additional comparisons with the main baseline GSBM
> on high-dimensional spheres with potential $U_\text{smooth}(x)$ and
> $\tau=0.01$. We use the Optimality and L2UV metric (from A. Korotin et. al  "Do Neural Optimal Transport Solvers Work?".) instead of Feasibility.
>
> | Data dim         | 10   | 100  | 500  | 1000  |
> |------------------|------|------|------|-------|
> | HOTA L2UV        | **0.189**| **0.034**| **0.059**| **0.109** |
> | HOTA Optimality  | **3.37** | **3.16** | **3.78** | **3.17**  |
> | GSBM L2UV        | 0.31| 0.136 |  0.110  |  0.142   |
> | GSBM Optimality  | 4.04  | 6.27  | 20.88 | 22.1  |
>
> Lastly, training on large-scale image datasets is non-trivial and
> requires further investigation. We attribute this to the joint
> optimization of the control policy and potential function within a
> single model architecture, where learning one component may interfere
> with the other. We partially addressed this issue through dynamic
> gradient scaling, which significantly improved training stability.
> Nevertheless, we consider this challenge an important standalone
> research direction.
>
> **Questions**
>
> **Q1: The Sphere tasks** involve transporting distributions around
> hyperspheres in $\mathbb{R}^d$ (with $d \in \{3, 9, 27, ..., 729\}$),
> where the source and the target measures are Gaussian and projected onto
> the opposite poles. The potential $U(x)$ enforces manifold constraints.
> We consider sharp and smooth potentials:
> $$U_\text{sharp}(x) =  C_d \mathbf{I} (\|x\| < 1)$$ where $C_d > 0$ is a
> dimension-dependent constant, and
> $$U_\text{smooth}(x) =  C_d \, \text{sigmoid} ((1 - \|x\|) / \tau)$$
> Thus, transport trajectories should not enter the unit sphere and,
> ideally, lie on the sphere surface, approximating optimal path on a
> sphere.The sharp potential tests scalability under strict constraints,
> while the smooth variant evaluates sensitivity to the relaxation
> parameter $\tau$ (Figure 2.b).
>
> **Q2: Ablation which removes the EMA splitting of the objective.** We
> thank reviewer for valuable suggestion to ablate HOTA with and without
> EMA. The intuition behind introducing EMA is similar to those in
> Q-learning in RL, the target (EMA) network is used to stabilize the
> training process and prevent harmful feedback loops that can arise when
> the same network is used for both time and space derivatives.
>
> |       | Feasibility  |         |         | Optimality |         |         |
> |-------|:--------------------------------------:|:-------:|:-------:|:--------------------------:|:-------:|:-------:|
> |       | Stunnel                               | Vneck   | GMM     | Stunnel                    | Vneck   | GMM     |
> | HOTA  | **0.006**                            | **0.002** | **0.19** | **320.90**               | 115.09  | **80.44** |
> | HOTA w/o EMA | 0.018                            | 0.004   | 0.65    | 338.67                   | **109.25** | 97.30   |
>
> **Q3: Acceleration and replay buffer ablations for the BabyMaze
> dataset.** Most of the OT paths in the presented 2D datasets are
> piecewise straight. Therefore, using acceleration in the $L_1$ integral
> norm reduces the total cost. The optimality increases with
> $\lambda_a > 0.1$ paths begin to cross the zone of the potential.
>
> | Acc. coef. $\lambda_a$ | 0    | 0.05 | 0.1  | 0.2  | 0.5  |
> |------------------------|------|------|------|------|------|
> | Feasibility            | 0.010 | **0.004** | 0.005 | 0.008 | 0.007 |
> | Optimality             | 5.24 | **4.87** | 5.10 | 5.45 | 7.30 |
> | Straightness $\int \| a_t \| dt$ | 4.91 | 3.25 | 3.02 | **2.91** | 3.58 |
>
>  BabyMaze without replay buffer has feasibility $0.008$ and optimality $16.46$
> (with buffer $0.004$ and $4.87$). In this case most trajectories
> ignore the obstacle given by the potential $U(x)$. Hence, the ablation
> results are intuitive.
>
> **Q4: lines 89-92** $\mu(\cdot | x)$ is conditional map induced by the SDE with the
> *optimal* drift $v*$ that minimizes (2). Taking into account all
> $x \sim \alpha$, $\mu$ maps measure $\alpha$ into measure $\beta$, such
> that $E_\alpha \mu(B | x)  = \beta(B)$ for any Borel subset $B \in R^d$.
> Notation $c(x, \mu)$ is the generally used notation in weak OT (ref. N.
> Gozlan \"Kantorovich duality for general transport costs and
> applications.\"). The cost depends only on the initial point $x$ and the
> distribution $\mu$, not on specific paths, because we are optimizing
> over all possible paths in equation (2). We will add these details to
> the introduction.
>
> **Q5: NLSB** In the main text, we will elaborate that NLSB is Koshizuka and Sato 2024
> Neural Lagrangian Schrodinger Bridge baseline.
>
> **Q6: Gray values in Table 2** means that the method with corresponding
> configuration significantly diverge from the target distribution
> $\beta$. For this reason, we omit such comparisons with the others in
> the Optimality table.
>
> **Limitations:**
>
> **L1: Do derivatives $\partial_t s, \nabla_x s, \Delta_x s$ add
> computational cost?** This is quite unexpected, but thanks to *jax.jit*
> optimization and efficient implementation of the Laplacian, the
> calculation of the indicated derivatives takes approximately **the same
> time as the calculation of the function** $s(t, x)$ itself (with a
> difference of no more than 5\%). We have evaluated the computation
> time directly with MLP(512, 512, 512, 1) network and input $x$ with
> dimensions in the range $10^2, 10^4, 10^5$.
>
> We compared the training time with the baseline GSBM on the same
> hardware.
>
> | Datasets                  | 2D benchmark | 1000D opinion depolarization |
> |---------------------------|--------------|-------------------------------|
> | GSBM time                 | 18-45 min    | 870 min                     |
> | HOTA (ours) time          | 6-7 min      | 26 min                        |
>
> We propose to include these computation times in the main text.
>
> **L2: The theoretical guarantees** mentioned in line 48 of the
> introduction are detailed in Section 6 (Proof of Theorem 1) of our
> paper. The paper reformulates the GSB
> problem as a dual optimization problem using Kantorovich potentials. The
> dual objective (8) enforces marginal matching (feasibility)
> through the Kantorovich potential $s(1,x)$, while the HJB equation (9)
> ensures trajectory optimality by minimizing the combined kinetic and
> potential energy. The method works for non-smooth costs $U(x)$ and
> complex geometries, as the HJB framework does not require
> differentiability of $U(x)$. Besides this, the paper guarantees the
> uniqueness of the HJB solution (in the viscosity sense) under coercivity
> conditions (eq. 10-11).

---

> > ### Author Response · Authors · 2025-08-06
> >
> > Dear Reviewer,
> >
> > We apologize for having to resort to this reminder, but we really enjoyed the theoretical extension that you had proposed and were hoping you would be able to discuss it within the remaining two days. Please kindly let us know if you appreciate the derivation above or if you have any other questions.
> >
> > Sincerely,
> > Authors of Submission 24195

---

> ### Comment · Reviewer_yN65 · 2025-08-07
>
> Thanks for the detailed response and additional results!
> I appreciate the additional, impressive ablation studies to show the impact of these algorithmic choices.   In particular, putting a stop gradient on various components of HJB regularizers should have wider applicability.   I had also missed the additional experiments in the supplementary material (likely prepared after the initial deadline).
>
> > $\mu(\cdot | x)$ is conditional map induced by the SDE with the optimal drift $v*$ that minimizes (2). Taking into account all $x \sim \alpha$, $\mu$ maps measure $\alpha$ into measure $\beta$, such that $E_\alpha \mu(B | x) = \beta(B)$ for any Borel subset $B \in R^d$.
>
> I encourage the authors to include these details.    **Why would the optimal (weak) transport map take the form of an SDE?**
>
>
> >The distinction between unregularized OT and regularized OT is blurred in this work, where we invoke the unregularized dual potential perspective for optimizing over SDEs.
>
> My original concern was related to this transition between (i) c-convexity statements of the dual problem and (ii) the Lagrangian formulation of (optimal? stochastic?) transport problems.
>
> For example, in the stochastic case, I might attribute the $\frac{1}{2}\|v_t\|^2$ term to KL divergence regularization with a zero-drift Brownian motion (via Girsanov theorem), whereas $U_t(x)$ plays the role of a 'stochastic-Lagrangian' cost.    I say 'stochastic-Lagrangian' cost to distinguish from the 'dynamical OT' cost that would appear in Benamou-Brenier or a 'Lagrangian cost' appearing in Eq. 2.
>
>
> > the calculation of the indicated derivatives takes approximately the same time as the calculation of the function
>  itself (with a difference of no more than 5%).
>
> This is an interesting point to include in the main text, to preclude similar concerns from interested readers / practitioners.
>
> > I cannot find the theoretical guarantees claimed in line 48
>
> I think I expect the term "theoretical guarantees" to refer to an algorithm rather than an optimality statement, so I would soften this statement unless it refers to the convergence/optimality/etc guarantees of a particular algorithm.

---

> ### Author Response · Authors · 2025-08-08
>
> Thank you for acknowledging our effort and for the additional insights. As suggested, we commit to include the clarifications about the stochastic mapping $\\mu(\\cdot|x)$, as well as a note about the correspondence between stochastic OT and density regularization (with a reference to article [1] Neklyudov et. al 2024 "Wasserstein Lagrangian Flows"); we will also discuss the influence of the stop gradients and  of the target network, the low overhead costs of calculating derivatives, and a more specific description of the ``theoretical guarantees''.
>
> **Transition between (i) c-convexity statements of the dual problem and (ii) the Lagrangian formulation of (optimal? stochastic?) transport problems**
>
> We will soften our statements in the introduction which primarily serve to provide intuition for deriving the dual problem. Yet, the proof of Theorem 1 establishes a formal connection between $c$-convexity in the dual Kantorovich problem and the Lagrangian formulation of stochastic transport. Here is how the transition works formally.
>
> One starts with the primal GSB problem (with $L$ representing Lagrangian functional):
> $$
> \\inf_v  E  [ \\int_0^1 L(t,x_t,v_t) dt  ] \\quad \\text{s.t.} \\quad x_0 \\sim \\alpha, \\, x_1 \\sim \\beta.
> $$
> Using Lagrange duality, the constraint $x_1 \\sim \\beta$ is enforced via a potential $g(y)$:
> $$
> \\inf_v \\sup_g  \\{  E  [ \\int_0^1 L \\, dt - g(x_1)  ] +  E_\\beta[g(y)]  \\}.
> $$
> Swapping $\\inf$ and $\\sup$ (under the regularity conditions) yields the dual problem:
> $$
> \\sup_g  \\{  E_\\alpha[g^c(x)] +  E_\\beta[g(y)]  \\}, \\quad \\text{where} \\quad g^c(x) = \\inf_v  E  [ \\int_0^1 L \\, dt - g(x_1) \\mid x_0 = x  ].
> $$
> Here, $g^c(x)$ is the stochastic $c$-transform, which directly relates to the notion of $c$-convexity.
> This framework is also formally related to weak optimal transport **if** we define the cost function as
> $$
> c(x, \\mu) = \\inf_v  E  [ \\int_0^1 L \\, dt  \\mid x_0 = x, \\, x_1 \\sim \\mu  ].
> $$
> In this case, it holds that
> $$
> g^c(x) = \\inf_{\\mu \\in \\mathcal{P}(Y)}  [c(x, \\mu) -  E _\\mu g(y)].
> $$
> We propose to add this clarification to the introduction and briefly summarize the main idea of the proof, stressing that the reference to the weak OT is intentional.
>
> **Why would the optimal transport map take the form of an SDE?**
> To derive the solution for $g^c(x)$, one may consider the expectation over **general** stochastic processes (not necessarily diffusion or one driven by an SDE).
> In a general case, we define the value function as:
> $$
> V(t, x) = \\inf_{v_{[t,1]}}  E  [ \\int_t^1 L(s, x_s, v_s) \\, ds - g(x_1) \\,\\bigg|\\, x_t = x  ],
> $$
> with a terminal condition $V(1, x) = -g(x)$. Then, $g^c(x) = V(0, x)$.
> Using the Dynamic Programming Principle, one can obtain the HJB Equation
> $$
> -\\partial_t V(t, x) = \\inf_{v}  \\{  E  L(t, x, v) + \\mathcal{A}^v V(t, x)  \\},
> $$
> where $\\mathcal{A}^v$ is the infinitesimal generator of the process:
> $$
> \\mathcal{A}^v V(t, x) = \\lim_{h \\to 0} \\frac{ E [V(t+h, x_{t+h}) - V(t, x) | x_t = x, v_t = v]}{h},
> $$
> and the optimal control satisfies:
> $$
> v^* = \\arg\min_v  \\{  E  L(t, x, v) + \\mathcal{A}^v V(t, x)  \\}.
> $$
> At this step, we need to decide on the type of the random process $x_t$. For diffusion (in practice, we often assume that $dx_t = v_t \\, dt + \\sigma \\, dW_t$):
> $$
> \\mathcal{A}^v V = v \\cdot \\nabla_x V + \\frac{1}{2} \\text{tr}(\\sigma \\sigma^T \\nabla_x^2 V).
> $$
> If the control $v_t$ is given by a general function $v_t = f(x_t, \\varepsilon_t)$, with $\\varepsilon_t$ representing external randomness, e.g., noise, the HJB equation becomes:
> $$
> -\\partial_t V(t, x) = \\inf_{f}  \\{  E _{\\varepsilon_t}  [ L(t, x, f(x, \\varepsilon_t))  ] +  E _{\\varepsilon_t}[ f(x_t, \\varepsilon_t)] \\cdot \\nabla_x V  \\}.
> $$
>
> To summarize, we propose to clarify in the revised text that the diffusion-based solution is a special case motivated by the tractability in optimizing the value function. The SDE structure is particularly advantageous in numerical implementations, as it allows efficient gradient-based optimization and aligns with existing literature on diffusion-based generative models.

---

### Note · Authors · 2025-08-12

We thank the reviewers for their constructive feedback and the chance to summarize our responses. We believe we have addressed all the questions on the theoretical formulation and novelty, supported HOTA’s scalability and computational efficiency, and clarified the experimental scope.  Below, we highlight the key improvements made in response to the reviewers comments:

1. **Robust experimental validation across scales.** Our comprehensive evaluation strategy spans:
  - *Foundational 2D benchmarks*: We have used established GSB test cases that enable clear visualization and direct comparison with prior work;
  - *Scalability*: HOTA outperforms existing methods (DeepGSB, GSBM), including on the 1000D opinion depolarization task (see supplement). Evaluations across dimensions (10–1000D) on HD Sphered datasets show consistent superiority over GSBM, for both Gaussian and complex distributions.
  - *Future work*: While initial image experiments revealed optimization challenges, our method holds notable promise to function in generative AI on the image data. Optimization schemes, including gradient scaling, EMA/target networks, etc., are a clear path forward and an exciting direction for future research.

2. **Demonstrated scalability and runtime efficiency.**
   Remark: The method’s scalability was questioned due to simulation requirements and all the second-order derivatives.
   Resolution:
     - Empirical results showed near-linear scaling with dimensionality (1k–9k) and simulation steps (10–50);
     - Derivative calculations add minimal overhead (~5%);
     - HOTA achieves significant speedups over a simulation-free baseline GSBM due to a simpler optimization objective and better dimensional scalability.

3. **Non-Differentiable Potentials.** Theoretical and experimental results prove HOTA's robustness to obstacle potential smoothness, effectively handling non-differentiable potentials with ease. This  enables novel applications across multiple domains, including metric learning and LLMs (support discrete token spaces through non-differentiable potential handling).

4. **Connection between stochastic and entropy-regularized OT.** As an interesting theoretical addition, suggested by R1, we have derived that the HJB equation absorbs noise effects into potential energy term via the proposed re-parameterization, explicitly linking diffusion regularization to deterministic transport maps. This will be included and supported by the proper references.

---

### Decision · Program_Chairs · 2025-09-17

**Decision:**

Reject

**Comment:**

This paper suggests an algorithm for the Generalized Schrodinger Bridge (GSB) problem combining the dynamic formulation with the dual one. The suggested algorithm optimizes the value function of the control problem by minimizing the residual loss of the  Hamilton-Jacobi-Bellman (HJB) PDE together with the dual potential loss.

During initial reviews the reviewers were not convinced by the significance of this method due to its limited experimental setup where both scalability to higher dimensions and in terms of performance and training cost, as well as the sheer number and complexity of examples were lacking. There were also questions about theoretical contributions and exposition. During rebuttal the authors provided many new results alleviating to an extent the scalability concerns and providing comparison to the main baseline for the GSB problem (GSBM) demonstrating advantages of their own method. Due to that some reviewers increased the score leading to a weak accept situation.

===

As recently advised by legal counsel, the NeurIPS Foundation is unable to provide services, including the publication of academic articles, involving the technology sector of the Russian Federation’s economy under a sanction order laid out in Executive Order (E.O.) 14024.

Based upon a manual review of institutions, one or more of the authors listed on this paper submission has ties to organizations listed in E.O. 14024. As a result this paper has been identified as falling under this requirement and therefore must not be accepted under E.O. 14024.

This decision may be revisited if all authors on this paper can provide proof that their institutions are not listed under E.O. 14024 to the NeurIPS PC and legal teams before October 2, 2025. Final decisions will be communicated soon after October 2nd. Appeals may be directed to pc2025@neurips.cc.